# Toddlers strategically adapt their information search

Francesco Poli [1,2] ✉, Yi-Lin Li[3], Pravallika Naidu[3], Rogier B. Mars [1], Sabine Hunnius[1] & Azzurra Ruggeri [4,5,6] ✉

Adaptive information seeking is essential for humans to effectively navigate complex and dynamic environments. Here, we developed a gaze-contingent eye-tracking paradigm to examine the early emergence of adaptive information-seeking. Toddlers ($N = 60$, 18-36 months) and adults ($N = 42$) either learnt that an animal was equally likely to be found in any of four available locations, or that it was most likely to be found in one particular location. Afterwards, they were given control of a torchlight, which they could move with their eyes to explore the otherwise pitch-black task environment. Eye-movement data and Markov models show that, from 24 months of age, toddlers become more exploratory than adults, and start adapting their exploratory strategies to the information structure of the task. These results show that toddlers' search strategies are more sophisticated than previously thought, and identify the unique features that distinguish their information search from adults'.

Humans' ability to explore is arguably (still) superior to any other form of artificial[1–3] or natural[4] intelligence. What confers an advantage to human exploration may be the ability to entertain a multitude of exploratory strategies and select the one that promises to maximize learning given the specific situation presented[5,6]—which has been referred to as ecological active learning competence[7]. This ability is fundamental to acquire information efficiently and effectively[8,9], to build complex systems of knowledge[10,11], and to shape the world around us in innovative ways[12]. Adaptive information-seeking is especially crucial during the first years of life, when children know the least and need to learn the most. However, it is still unknown whether the drastic learning that occurs in the very early stages of life is supported by this adaptive information-seeking competence.

Current evidence on whether adaptive information search is already present from early in life is conflicting. Some studies have shown that young children cannot optimally select the best exploration strategy when multiple options are available[13,14]. For example, when given a clue that could narrow their search (e.g., empty cup), toddlers do not preferentially choose the one cup that offers a guaranteed reward[14]. However, other studies have documented early signs of adaptive search in infants and toddlers. For example, infants are more likely to solicit information from a knowledgeable adult compared to an ignorant or unreliable one[15,16]. Similarly, they rely on social partners when presented with cognitively demanding tasks, but tackle them on their own otherwise[17]. After failing to activate a toy, 16-month-old infants who were made to believe that the failure was due to their own inability were more likely to seek for help, while infants who were made to believe that the toy was malfunctioning were more likely to test the same behavior on other toys[18]. Relatedly, after observing an object unexpectedly pass through a wall, infants engaged in behaviors directed at testing their solidity (i.e., banging the object on a table)[19]. Taken together, these studies show that early search behavior is not rigid, as different environments elicit different responses. However, these between-subjects designs do not examine whether toddlers can dynamically adapt to changes in the environment, flexibly switching between different search strategies depending on the specific characteristics of the task at hand.

[1]Donders Institute for Brain, Cognition and Behaviour, Radboud University, Nijmegen, Netherlands. [2]MRC Cognition and Brain Sciences Unit, University of Cambridge, Cambridge, UK. [3]Wellcome Centre for Integrative Neuroimaging, Centre for Functional MRI of the Brain (FMRIB), Nuffield Department of Clinical Neurosciences, John Radcliffe Hospital, University of Oxford, Oxford, UK. [4]Max Planck Research Group iSearch, Max Planck Institute for Human Development, Berlin, Germany. [5]School of Social Sciences and Technology, Department of Education, Technical University Munich, Munich, Germany. [6]Department of Cognitive Science, Central European University, Vienna, Austria. ✉e-mail: francesco.poli@mrc-cbu.cam.ac.uk; a.ruggeri@tum.de

So far, this adaptive competence has been investigated only in older children and adults, relying on tasks that require participants to tailor their search strategies to the changing characteristics of the environment[7]. For example, information search can target specific hypotheses (e.g., "Is it the penguin?" when trying to find out what animal can endure the lowest temperatures) or narrow down the range of hypotheses under consideration (e.g., by asking "Does it have wings?"). From 3 years of age, children flexibly switch between different types of questions depending on the context[5,20], indicating that they are indeed able to adjust their exploration strategies to the statistical structure of a task.

In this paper, we introduce an experimental paradigm that allows us to investigate how toddlers adapt their exploration strategies to the characteristics of given environments. We devised a gaze-contingent

eye-tracking task that allowed toddlers between 18 and 36 months of age, as well as adults, to actively and dynamically explore the environment presented on the screen (Fig. 1A). Although previous studies have already exploited gaze-contingent paradigms to probe infants' cognitive abilities with minimal task and verbal demands[21,22], this study allows toddlers to actively and independently control their exploration. During training (Fig. 1B), participants either learnt that an animal was equally likely to be found in any of four available locations (Uniform condition), or that it was most likely to be found in one particular location (Skewed condition). At test, they were given control of a gaze-contingent "torchlight", which they could move with their eyes to actively explore the otherwise pitch-black task environment to find the hidden animal (Fig. 1C). An informative cue (i.e., a treasure chest) reliably predicted the location of the target animal. In the Uniform

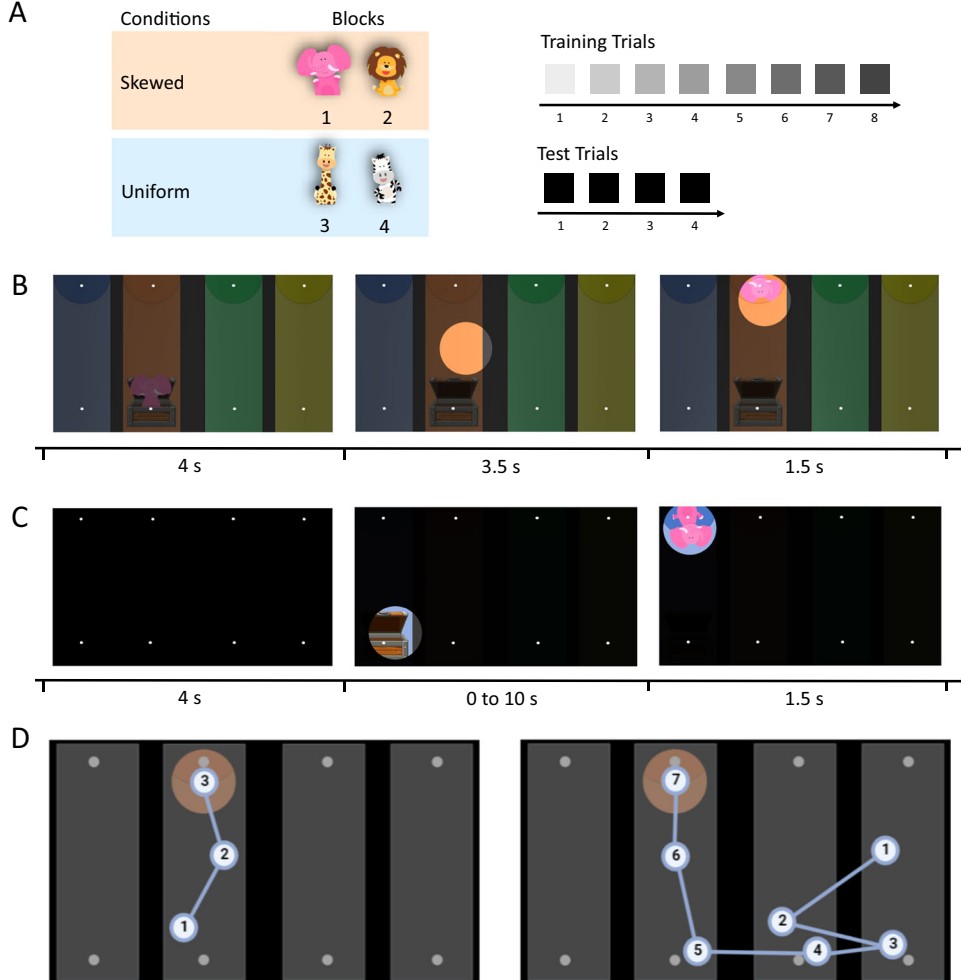

**Fig. 1 | The gaze-contingent torchlight task. A** Participants were presented with two within-subject conditions, Skewed and Uniform, each consisting of two blocks (numbered 1–4) composed of training and test trials. Each trial had a specific level of darkness (a qualitative indication is reported by the squares on the right). **B** During the training phase, participants observed 8 trials where an animal jumped out of a treasure chest, moved upward, and disappeared at the top of the screen in one of four differently colored areas. The animal then briefly reappeared at its disappearance location. Throughout training, the screen gradually became darker to familiarize participants with the gaze-contingent torchlight (the lighter circle shows an example of the torchlight). In the Skewed condition, the animal consistently appeared in the same colored area. In the Uniform condition, the animal appeared in a different colored area each time. In both conditions, the treasure chest always reliably indicated the target location. **C** At test, the screen was completely dark, and participants could only hear the animal jumping out of the treasure chest. They were then given up to 10 s to search for the target animal using

the torchlight (i.e., the search phase). Since they knew the animal appeared only briefly, participants were motivated to find its hiding location before it appeared, so as not to miss it. In the Skewed condition, the animal appeared as soon as participants identified its location; in the Uniform condition, the animal appeared only after participants had found the cue (i.e., the treasure chest) and used that information to identify the correct target location. **D** Examples of the most efficient search in the Skewed (left) and Uniform (right) conditions. In the Skewed condition, the first look is already located within the correct portion of the screen, indicating that the participant exploited the information structure acquired during familiarization to correctly anticipate the target location. In the Uniform condition, the participant engages in more scanning behavior to find the cue, which informs them about the location of the target. [Treasure chest images with copyright from Babysofja via Creative Market and animal images freely available from Prora via Pixabay].

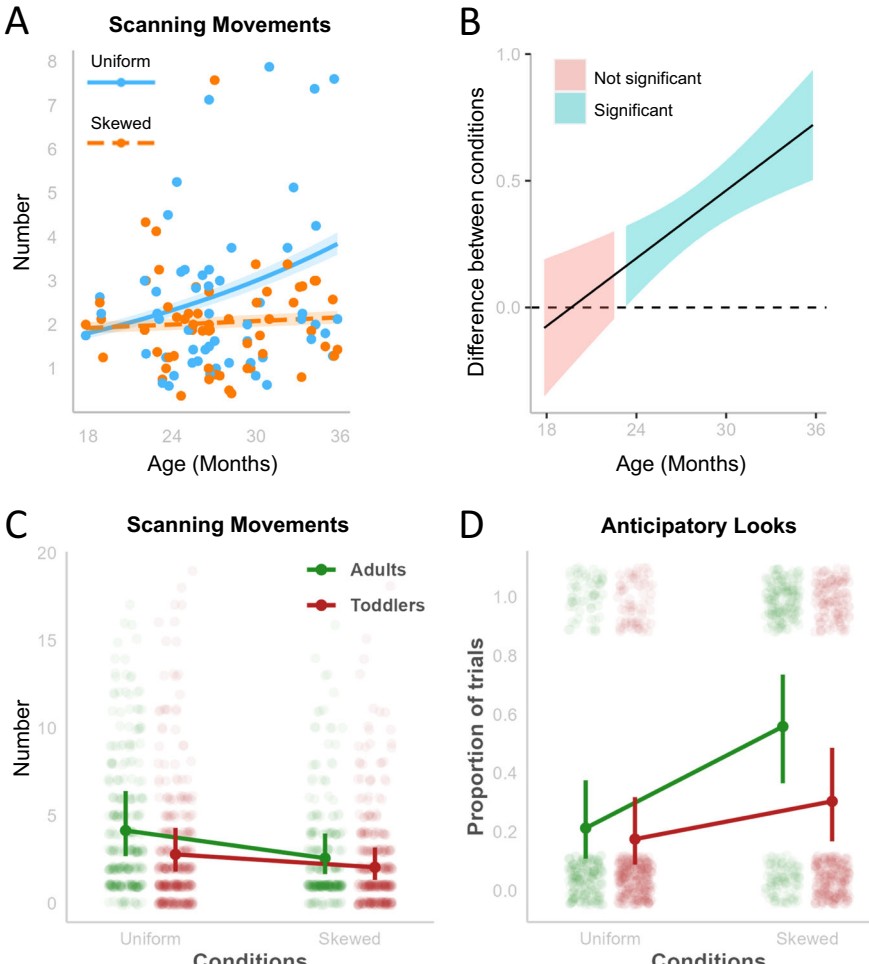

**Fig. 2 | Scanning movements and anticipatory looks in toddlers and adults.**
**A** Predictive estimates of the number of scanning movements across age show an interaction between age and condition. Dots indicate the mean number of trials for each toddler, shaded areas indicate the standard error. **B** Johnson-Neyman intervals show that the difference between conditions is significant from 24 months of age onward. **C** Predictive means and standard errors for the number of scanning movements show that both adults and toddlers performed more scanning movements in the Uniform condition compared to the Skewed condition. Points show the distribution of the data (jittered). Vertical lines indicate the standard error. **D** Predictive means and standard errors for the percentage of correct anticipatory looks show that both adults and toddlers produced more anticipatory looks in the Skewed condition compared to the Uniform condition. Points show the distribution of the data (jittered). Vertical lines indicate the standard error.

condition, participants did not know where the animal was hidden. Hence, the most efficient search strategy consisted in scanning across the different locations to find the cue, and in using this information to constrain their subsequent search and identify the correct target location (Fig. 1D). In the Skewed condition, where the animal always hid in the same location on all trials, participants had collected sufficient statistical information to make an informed prediction about the most likely target location, and could direct their information search accordingly.

Here, we find that both toddlers' and adults' exploratory patterns varied across conditions, thus demonstrating adaptive exploration skills. Specifically, participants in the Uniform condition engaged in more visual scanning to find the informative cue, and devised more complex search patterns (as predicted by a reinforcement-learning model, see "Results" section). Conversely, in the Skewed condition, participants produced fewer scanning eye-movements and adopted simpler search patterns. Also, in the Skewed condition, participants engaged more in anticipatory looks to the most likely cue and target location, by gazing towards the correct portion of the screen even before the search phase had started. Overall, this paradigm allowed us to examine 18- to 36-month-old toddlers' active search for information, capturing the emergence of active and adaptive exploration and

tracing its early development into the advanced search strategies that we can observe later in childhood and adulthood.

## Results
### Toddlers and adults adapt their search to the information structure of the environment

Both toddlers and adults performed the task successfully, correctly identifying the location of the hidden animal in both conditions (see "Performance" in the "Methods" section). Crucially, they adapted their exploratory strategy to the different information structures of the task (i.e., the different likelihood distributions associated with the Skewed and Uniform conditions). In particular, as age increased, toddlers engaged in more scanning behavior, searching more for the cue in the Uniform compared to the Skewed condition ($z = 3.29$, $\beta = 0.04$, 95% CI [0.01, 0.06], $p < 0.001$) (Fig. 2A). The difference between Uniform and Skewed conditions in the number of scanning movements emerged from approximately 24 months of age (24.08 months, $t(748) = 2.67$, $\beta = 0.20$, 95% CI [0.05, 0.34], $p = 0.008$) (Fig. 2B), indicating that toddlers were seeking information adaptively from 2 years of age onward.

When comparing toddlers' and adults' scanning behavior (Fig. 2C), we found that adults performed more scanning movements than toddlers in both conditions (Uniform: $t(1334) = 6.60$, $\beta = 0.38$,

SE = 0.06, $p < 0.001$; Skewed: $t(1334) = 3.10$, $\beta = 0.21$, SE = 0.07, $p = 0.011$), but both groups showed a comparable pattern of results with more scanning movements in the Uniform condition ($z = 9.59$, $\beta = 0.48$, 95% CI [0.38, 0.58], $p < 0.001$). This replicates previous evidence reporting adaptive behavior in school-age children and adults[9], and validates our measure of adaptive behavior, thus strengthening the results obtained with toddlers.

Following previous work[23], we analyzed whether participants performed anticipatory looks (i.e., looks before the start of the search phase) onto the correct portion of the screen (i.e., column). Since model comparison showed that toddlers' age (in months) did not improve model fit (see "Methods" section), we directly compared toddlers' and adults' anticipatory looks across conditions (Fig. 2D). Adults and toddlers showed a comparable pattern of results, with more anticipatory looks in the Skewed compared to the Uniform condition (Adults: $t(1459) = -8.09$, $\beta = -1.55$, SE = 0.19, $p < 0.001$; Toddlers: $t(1459) = -4.28$, $\beta = -0.72$, SE = 0.17, $p < 0.001$).

Taken together, these findings indicate that both toddlers and adults adapted their visual search strategies to the information structure of the search space. However, the increased scanning behavior in the Uniform condition might reflect random search driven by a general state of confusion, rather than an adaptive response. We addressed this concern by using Markov models, which allowed us to test participants' performance against the most effective strategy and random behavior.

## Toddlers and adults devise more complex exploratory patterns when the environment is less predictable

To test whether participants learned to flexibly adjust their exploration across conditions, we pitted participants' exploratory eye-movements against the behavior of a reinforcement learning model. Specifically, in two sets of simulations, the model was introduced to either the Skewed or the Uniform condition, and learned how to efficiently search the target through trial and error. These simulations (see "Methods" section and Fig. 3c) indicated that more predictable environments (i.e., the Skewed condition) are easier to learn and should result in simpler patterns of information search, as the location of the target is fully predictable without the aid of the cue. Conversely, less predictable environments (i.e., the Uniform condition) are harder to learn, and call for more complex patterns of information search, because they require to engage in a broader exploration directed at finding the cue. We additionally defined a random search pattern, which consists of stochastic eye-movements that are completely independent of where the cue and the target are located. Hence, these movements result in random transitions from any location to any other in a completely unpredictable manner, thus corresponding to the highest possible level of complexity.

After simulating how efficient search patterns can be learned (as well as a random search pattern) with a reinforcement-learning model, we estimated the complexity of these patterns using Markov models. Markov models allow us to compute a transitional probability matrix which specifies the probability to go from a given location (i.e., state) on the screen to any other location (Fig. 3A, B; see "Methods" section for details). More complex exploratory patterns result is more complex transitional probability matrices, as indexed by their entropy. Indeed, we find that random search is the most complex (log-entropy = 2.77), followed by the most efficient search for the uniform condition (log-entropy = 2.46), while efficient search for the Skewed condition is the least complex (log-entropy = 2.34).

After computing the complexity of the most efficient search patterns and the random search pattern, we fitted two distinct Markov models for Skewed and Uniform conditions, separately for three age groups: toddlers below 24 months of age (for whom we did not find statistically significant evidence of adaptive behavior from the behavioral analyses, $N = 14$), toddlers above 24 months of age ($N = 46$), and adults ($N = 42$). We obtained six transitional probability matrices that specified the exploratory patterns of each condition for each age group (Fig. 3B). To obtain a measure of the complexity of the exploratory patterns, we computed the entropy of the transitional probability matrices.

We compared participants' performance to the most efficient and random search patterns (Fig. 3D). For the Skewed condition, we found that the exploratory patterns of both adults and older toddlers were more complex than required by the most efficient search (model log-entropy = 2.34; adults: mean log-entropy = 2.413, 95% CI = [2.403, 2.423]; toddlers: mean log-entropy = 2.455, 95% CI = [2.446, 2.465]). Similarly, for the Uniform condition, the complexity of the exploratory pattern of adults and older toddlers were more complex than required by the most efficient search (model log-entropy: 2.46, adults: mean log-entropy = 2.485, 95% CI = [2.475, 2.495], toddlers: mean log-entropy = 2.524, 95% CI = [2.514, 2.534]). In all cases, participants' patterns were far more systematic than random search (log-entropy = 2.77).

We found that both adults and older toddlers produced more complex patterns when searching for information in the Uniform compared to the Skewed condition (adults: $z = 10.32$, $\beta = 0.07$, SE = 0.01, $p < 0.001$; older toddlers: $z = 9.83$, $\beta = 0.07$, SE = 0.01, $p < 0.001$), while younger toddlers showed the opposite effect ($z = -3.51$, $\beta = -0.02$, SE = 0.01, $p = 0.006$) and overall produced far simpler exploration patterns than adults ($z = 39.69$, $\beta = 0.27$, SE = 0.01, $p < 0.001$) and older toddlers ($z = 45.26$, $\beta = 0.32$, SE = 0.01, $p < 0.001$). Although the difference between conditions was of the same magnitude for older toddlers and adults, older toddlers produced more complex patterns than adults in both conditions (Uniform: $z = 5.57$, $\beta = 0.04$, SE = 0.01, $p < 0.001$; Skewed: $z = 6.07$, $\beta = 0.04$, SE = 0.01, $p < 0.001$).

Finally, we tested for qualitative differences between adults' and toddlers' search patterns. Specifically, we analyzed the differences in transitional probability matrices between adults and toddlers with a logistic regression (Fig. 4). This allowed us to show that toddlers displayed an increased exploration of the cue locations compared to adults ($z = 2.86$, $\beta = 1.66$, SE = 0.58, $p = 0.004$), indicating an enhanced tendency to seek information.

Overall, these results show that, from 24 months of age, toddlers increase the complexity of their exploration search when the environment is less predictable (i.e., in the Uniform condition)—just like adults do—thus displaying adaptive search behavior. At the same time, they devise more exploratory patterns than adults, systematically targeting their exploration toward the relevant portions of the screen (i.e., the cue locations). This shows the unique traits that distinguish toddlers' information search from adults'.

## Discussion

Adaptiveness is a fundamental aspect of human exploratory behavior[24]. It allows us to tailor our information-seeking and learning strategies to the ever-changing constraints and characteristics of the world[7,25], maximizing learning success and effectiveness while optimizing the deployment of cognitive resources[26]. In the current work, we show how the ability to seek information adaptively emerges across the first years of life. We used a gaze-contingent task in which 18- to 36-month-old toddlers and adults controlled a torchlight using their gaze. Compared to previous studies[20], this paradigm allowed us to test active learning abilities in much younger children, and to look at more fine-grained, model-based measures of information search. We found that active information search is already successful at 18 months of age, as indicated by toddlers' ability to find the target by exploiting the informative cue. Moreover, we found that adaptiveness in toddlers' information search emerged at the age of 24 months, as suggested by the increase in scanning behavior in the Uniform condition, and the increase in anticipatory looks in the Skewed condition.

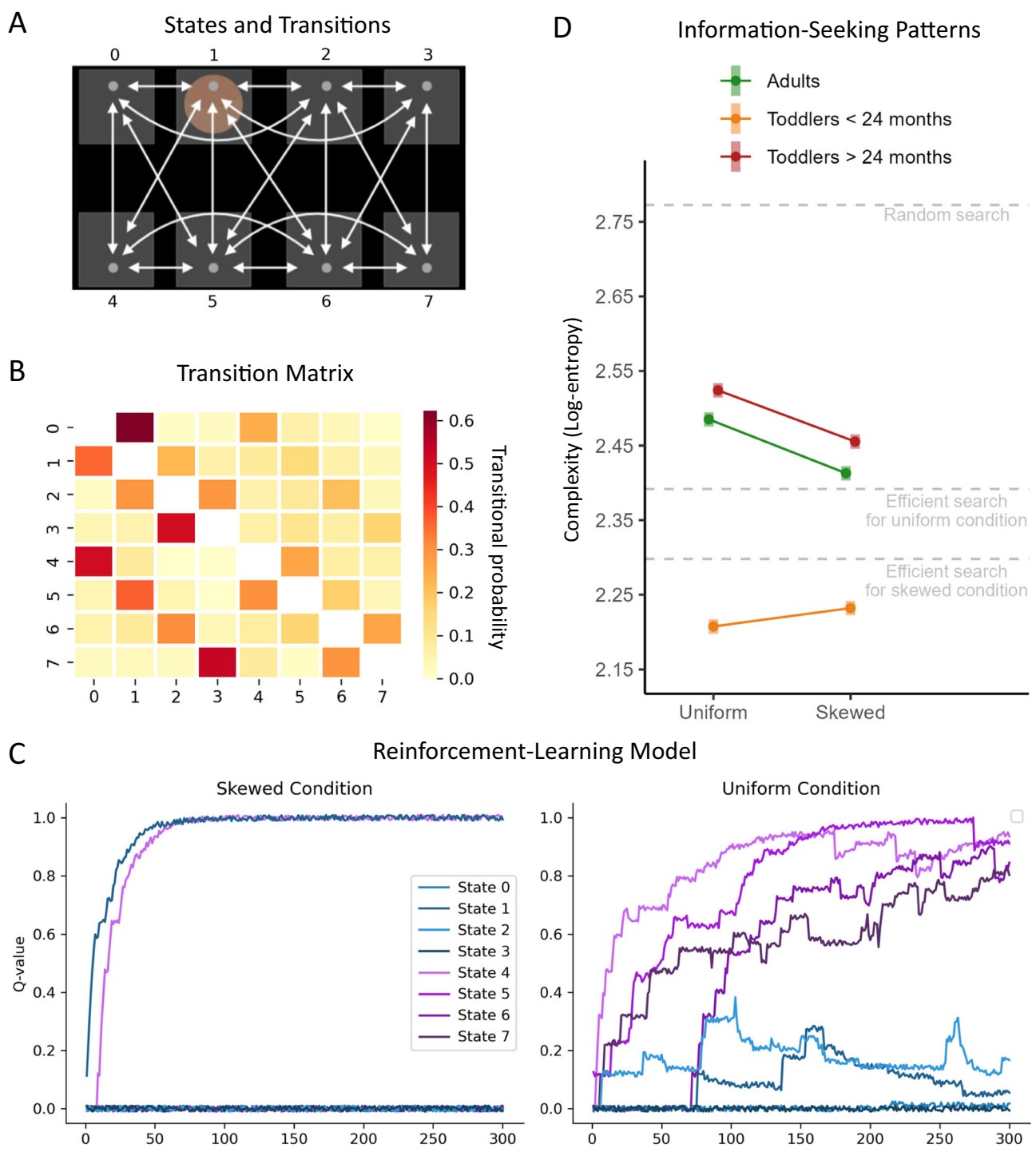

**Fig. 3 | Results from the Markov models. A** The Markov model computes the probability of transitioning from each state (i.e., the 8 locations on the screen, indicated here with numbers from 0 to 7) to any other state. **B** The transition matrix is the output of the Markov model, after observing the data. The data was divided by condition (Skewed vs. Uniform) and age (younger toddlers, older toddlers, and adults), resulting in six transition matrices. A measure of entropy was computed for each matrix, thus quantifying the complexity of the exploratory patterns. **C** To better assess participants' performance, we compared it to a reinforcement-learning model that learned to find the target animal in the Skewed and Uniform conditions. In the Skewed condition, the model learned that locations close in space to the target location (e.g., locations 1 and 4, when the target appears in location 0) are valuable, as they directly lead to the target. In the Uniform condition, the model learned that the cue locations (in purple) are valuable, as they might contain information that leads to the target. **D** Complexity of the exploration patterns for participants (in colours) and for the reinforcement-learning model (dashed grey lines). Complexity was higher for the Uniform compared to the Skewed condition for adults and older toddlers, but not for younger toddlers. Shaded areas indicate the standard error.

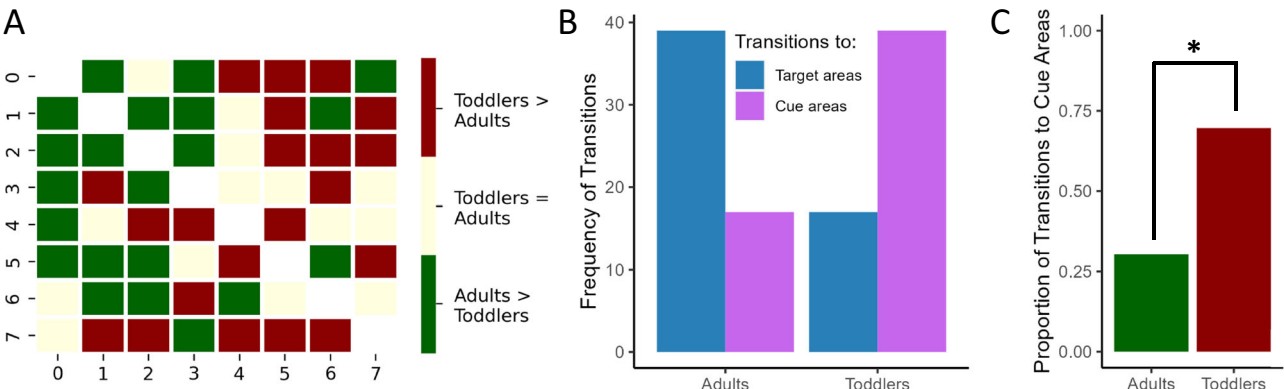

**Fig. 4 | Differences between toddlers' and adults' search patterns.**
**A** Comparison between the transition matrices of adults and toddlers, where red indicates transitions (i.e., eye movements from one location to the other) that were more frequent in toddlers, and green indicates transitions that were more frequent in adults. **B** The overall frequencies of transitions that were more likely to be performed by toddlers and adults. **C** A logistic regression showed that adults were more exploitative (i.e., more likely to make a transition to a target area), while toddlers were more exploratory (i.e., more likely to make a transition to the cue areas). * indicates a *p*-value < 0.001 for a two-tailed *z* test.

Previous research indicates that children under the age of 3 struggle to effectively narrow down their search for a hidden reward among multiple options[13,14]. For instance, in a recent study[14], toddlers were presented with four cups (A, B, C, and D) arranged in two pairs (A and B versus C and D). One cup in each pair contained a hidden sticker, but the specific cup was unknown. When one cup was revealed to be empty (e.g., cup B), toddlers should have realized that finding a sticker in cup A was certain, while finding one in cup C or D was only possible. However, their behavior did not demonstrate a preference for cup A. Although similar in structure to our task, these studies differed in two important ways. First, success in previous tasks required logical abilities that relied on abstract, combinatorial thought, while the current task required probabilistic reasoning about statistical regularities—an ability that emerges earlier in infancy[27]; Second, the current task does not involve any explicit choice or action planning besides eye movements, while previous studies required both. These simplifications allowed us to capture the early emergence of adaptive information seeking, tapping as little as possible into other competences.

Although previous studies identified simpler forms of active learning from early infancy[16,28–31], the developmental change in adaptive information-seeking across the first years of life was still poorly understood. By examining a crucial period in children's cognitive development, the current research not only reveals toddlers' advanced exploratory strategies, but also the dynamics of their developmental change. This developmental change might depend on multiple factors. First, recent findings show that infants possess the ability to exploit past experiences to discover better learning strategies[32,33]. After 8-month-old infants learned that multiple environments shared the same probabilistic structure, they were able to rely on this knowledge to quickly identify informative stimuli in new environments[32]. In this sense, adaptive information-seeking competence might be gradually learned after failing with simpler, more rigid exploration strategies, such as random or hypothesis-probing exploration. Second, cognitive flexibility is still drastically developing across toddlerhood[34,35]. A key challenge for younger children is resolving how to respond flexibly when a task presents conflicting information, such as conflicting rules or bivalent stimuli[35]. These skills develop across the second and third year of life[36], and they might be at the foundation of toddlers' emerging ability to flexibly switch information-seeking strategy based on the different probabilistic structure of each environment.

Finally, the results of the Markov models show that, by 24 months of age, toddlers demonstrate a sophisticated, adult-like ability to search for information adaptively, but they also display unique learning characteristics. The exploration patterns detected by the Markov models indicate that 24-to-36-month-old toddlers' exploration was driven to the informative cue locations more so than adults' exploration, demonstrating that toddlers displayed a greater tendency towards gaining information. Moreover, toddlers increased the complexity of their exploratory patterns more than adults, and more than expected from the theoretical estimates of efficient behavior. Although this might be considered a bug or faulty process, it emerges as a feature when considering the ecological[37] and bounded[6,38] nature of the human mind. In fact, in toddlers' propensity to incur unnecessary costs might reside the unique opportunity to discover unexpected information and generate ex-novo solutions[39,40]. These results call for further research to test the distinctive drives of young children's exploration and hypothesis generation.

## Methods

### Participants

To compute the sample size, we simulated synthetic data using information from pilot data (*N* = 5) and data from a study with a similar design and population (*N* = 37), as well as theoretical constraints (code is available at https://osf.io/rfx5u/). We simulated the expected results 1000 times and identified that power above 80% was expected to be reached with a sample of toddlers ≥59 (Mean power = 86.60%, CI = [84.33, 88.65]) and a sample of adults ≥28 (mean power = 85.00%, CI = [82.63, 87.16]) due to reduced noise in adult data. Forty-two adults (*M* = 37.02 years, SD = 16.77, range = 18 − 73 years, F = 24, M = 18) with normal to corrected vision and no mobility disorders as prerequisites, as well as 60 toddlers (mean = 27.36 months, SD = 0.37, range = 17.8–35.8 months, F = 35, M = 25) were recruited from the online database of the Max Planck Institute for Human Development, Berlin, Germany. Informed consent was obtained from adults and from children's legal guardians prior to participation. Participating adults and families received €10 for their participation. The study was approved by the local ethical committee of the Max Planck Institute for Human Development, Berlin, Germany.

### Design

Participants were presented with two experimental conditions in a within-subjects design (order of conditions counterbalanced). Each condition included two blocks composed of eight training trials and four test trials (Fig. 1). In every trial (training and test), the screen was divided into four columns of different colors (blue, orange, green, and yellow), and a treasure chest was located at the bottom of one of the four columns. In the trials of the Skewed condition, the treasure chest was always located in the same column (although this location was

different in the two blocks). In the Uniform condition, the treasure chest was located in a different column on every trial, in a pseudo-randomized order. Each of the four blocks featured a different animal (elephant, giraffe, zebra, and lion). For an illustration of the paradigm, see Supplementary Movie 1.

## Procedure

Each trial started when participants first looked at the screen. In the training trials, the treasure chest opened producing a sound, an animal jumped out of the treasure chest accompanied by jumping noises, and then jumped all the way up, disappearing from the upper part of the screen. This hiding sequence all happened within the same column and lasted 4 s. After an inter-stimulus interval (ISI) of 3.5 s, the animal reappeared from the same location where it had disappeared, waved, and smiled, accompanied by a happy sound, before disappearing again (length of the reappearing sequence: 1.5 s). If participants were still looking at the screen after the end of the trial, a new trial was immediately played by the experimenter. Across the 8 training trials within a block, the luminosity of the screen diminished: The screen got darker trial after trial, although even in the last training trial all the elements and colors were still visible. On each trial, after the hiding sequence (i.e., when the animal disappeared), a circular area on the screen (diameter = 200px) turned to full luminosity, creating the impression of a torchlight. The movement of this torchlight was contingent to the gaze of the participants (i.e., the area of the screen where participants were foveating turned to maximal luminosity), and participants could move the torchlight around with their eyes to explore the screen. The torchlight remained on the screen during the ISI and during the reappearance sequence, until the end of the trial.

Test trials were identical to training trials, except for two key differences. First, the screen was now pitch black. Participants could still hear the sounds accompanying the hiding sequence (i.e., the treasure chest opening and the animal jumping), but they could not see anything. Second, the duration of the ISI was contingent to the participants' gaze. In the Skewed condition test trials, the reappearing sequence did not start until (and unless) participants fixated on the correct target location for a minimum of 200 ms. In the Uniform condition test trials, the reappearing sequence did not start until (and unless) participants fixated the correct target location after having first foveated on the cue (i.e., the treasure chest) for a minimum of 150 ms. With these criteria, we made sure that participants in the test trials of the Uniform condition found the target animal only by applying the correct exploratory strategy (i.e., finding the cue first), and not by chance. If the criteria were not met after 10 s, the sound associated with the reappearing sequence was played, but the sequence was not shown on the screen. In this way, we communicated to the participants that the trial was over, and that they were not successful in finding the animal.

During familiarization trials, the screen was always visible, although it became increasingly darker. Thus, it was always possible to track the location of the target animal without the need to rely on the informative cue. However, during test trials, where the screen was completely dark, participants could not rely on tracking the animal anymore. In the Skewed condition, they could use the previously-observed evidence to predict its most likely location. In the Uniform condition, finding the animal required scanning across cue locations.

## Familiarization and calibration

Before the study began, the experimenter engaged the participating toddlers in a short play session to familiarize them with the lab environment. Both toddlers and adults were positioned approximately 60 cm away from the screen. Toddlers were seated on the caregiver's lap or on a high chair, with the caregiver sitting next to them. A Tobii Eye-tracking device (model: IS4 Large peripheral) was positioned below a 24" screen, and a webcam was placed above the monitor to

record the sessions. Black curtains covered the table on which the monitor was placed, as well as the wall behind it.

First, participants performed a 3-point calibration. For toddlers, caregivers were instructed not to look at the screen. In case toddlers looked away from the screen during calibration, the experimenter guided their attention back to the screen. After calibration, the study started. During toddlers' first training trial of the first block, at the end of the hiding sequence (i.e., when the animal hid), the experiment said: "Where is the elephant? Can you search for it with your eyes?" and after a few trials, the experimenter said: "Oh, it gets darker, but you can use your eyes like a torchlight" or "Oh, it gets darker, but when you look you can lighten it up." The level of the experimenter's engagement depended on the child's attention and responsiveness. For adults, instructions were the following: "Once I start the game, you will see different animals jumping out from the treasure chests and hiding on the top of the screen. All you have to do is find these hidden animals."

The study was programmed in Python (version 3.6) and a connection to the eye-tracker was established via Tobii research (Tobii Pro SDK). Code for running the study and stimuli is available online: https://osf.io/rfx5u/.

## Analysis

**Model comparison.** Four dependent variables were extracted for each test trial from the raw eye-tracking data: (1) A yes-no variable specifying whether participants successfully found the target during the search phase of each trial (i.e., successful search, mean = 0.43); (2) The time (in seconds) that participants employed to correctly identify the target (mean = 3.09, range = [1.22, 10.49]); (3) The number of scanning eye-movements between the different columns during the search phase (i.e., scanning movements, mean = 2.72, range = [0; 19]); (4) A yes-no variable specifying whether participants were already looking at the correct column when the torchlight appeared on screen (i.e., anticipatory looks, mean = 0.30).

For the regression models, we used the *bam* function of the *mgcv* package in *R*. On toddlers' data, a comparison was performed between 4 models: (1) a model including only the main effect of experimental condition (Skewed vs Uniform); (2) a model additionally including the main effect of age; (3) a model additionally including the interaction between condition and age; (4) A null model with no predictors. All models included trial number, block number, subject number, and presentation order of the conditions as random effects. Successful search and anticipatory looks were analyzed with logistic models, time to the target was analyzed with linear models, and scanning movements were analyzed with Poisson models. The goodness of fit of the models was evaluated with AIC as it is reported in Table 1. Only the best-fitting model was reported and interpreted.

To compare toddlers' and adults' data, the regression model included the main effects of experimental condition (Skewed vs Uniform), age group (Toddlers vs Adults) and their interaction. Random effects for trial number, block number, participant number, and order of presentation of the conditions were added. This model was superior to null models including only the random effects (successful search: AIC = 1204 vs AIC = 1394; time to target: AIC = 1026 vs AIC = 1123; scanning movements: AIC = 6849 vs AIC = 7008; anticipatory looks: AIC = 1659 vs AIC = 1760).

Performance results (successful search and time to target) are reported below, while results for scanning movements and anticipatory looks are reported in the main paper. All tests were two-tailed. Data always met the assumptions of the statistical tests used.

**Performance.** The best-fitting model of toddlers' performance was generalized logistic regression where successful search on each test trial (yes/no) was predicted by condition (Skewed vs. Uniform) and age (in months), with participant number, trial number, block number, and

**Table 1 | AIC scores of the regression models for successful search and scanning movements**

| Model | Successful search | | Time to target | | Scanning movements | | Anticipatory looks | |
|---|---|---|---|---|---|---|---|---|
| | AIC | ΔAIC | AIC | ΔAIC | AIC | ΔAIC | AIC | ΔAIC |
| Null | 790 | | 608 | | 3917 | | 979 | |
| Condition | 756 | 34 | 593[a] | 15 | 3876 | 41 | 962[a] | 27 |
| Condition + age | 755[a] | 35 | 595 | 13 | 3864 | 53 | 966 | 23 |
| Condition[a] age | 757 | 33 | 597 | 11 | 3853[a] | 64 | 968 | 21 |

ΔAIC is computed as the difference from the null model.
[a]Indicates the best-fitting models.

condition order as random effects. We find that toddlers' performance was above zero in both conditions ($z = -4.68$, $\beta = -3.00$, 95% CI [−4.26, −1.74], $p < 0.001$), and that they performed better in the Skewed compared to the Uniform condition ($z = 5.79$, $\beta = 1.13$, 95% CI [0.75, 1.52], $p < 0.001$). There was no significant effect of age on performance ($z = 1.50$, $\beta = 0.03$, 95% CI [−0.01, 0.08], $p = 0.135$).

When comparing toddlers' and adults' performance across conditions, post-hoc tests (Tukey method) were implemented with the *emmeans* package in *R*. They showed that adults performed better than toddlers in both conditions (Uniform: $t(1361) = 12.20$, $\beta = 2.56$, SE = 0.21, $p < 0.001$; Skewed: $t(1361) = 13.48$, $\beta = 4.41$, SE = 0.33, $p < 0.001$), but they showed a comparable pattern of results, with better performance in the Skewed condition than in the Uniform condition ($t(1361) = -9.09$, $\beta = -3.01$, SE = 0.33, $p < 0.001$).

When analyzing the time employed to find the target, only trials in which the target was successfully found were included in this analysis. The model comparison showed that toddlers' age (in months) did not improve model fit (Table 1). Hence, we directly report the results of a generalized linear model with condition (Skewed vs. Uniform), age group (Toddlers vs. Adults), and their interaction as independent variables, and participant number, trial number, block number, and condition order as random effects. With post-hoc tests (Tukey method), we found that both toddlers ($t(563) = 4.48$, $\beta = 0.50$, SE = 0.11, $p < 0.001$) and adults ($t(563) = 7.18$, $\beta = 0.39$, SE = 0.05, $p < 0.001$) were faster in the Skewed condition.

**Reinforcement-learning model.** We used a reinforcement-learning model to simulate how the most efficient search patterns could be learned in the Skewed and Uniform conditions. We define $S$ as the set of 8 discrete spatial states on the screen (i.e., the locations, or AOIs), with the particular neighborhood structure reflecting the spatial configuration (Fig. 3A). Specifically, we denote $S = \{0, 1, 2, 3, 4, 5, 6, 7\}$. From each state $s$ in $S$, the set of available actions $A(s)$ corresponds to moving to a neighboring state. The neighbors for each state are predefined based on the spatial configuration.

The learning process is modeled through Q-Learning, utilizing a 3-dimensional Q-table $Q$, indexed by $s, a, c$ where $s$ is the state, $a$ is the action, and $c$ is the cue to the target. Assuming that the model is motivated by finding the target (i.e., the target is rewarding), the update rule for $Q$ is:

$$Q(s,a,c) = Q(s,a,c) + \alpha\left[r + \gamma \max_{a'} Q(s',a',c) - Q(s,a,c)\right] \quad (1)$$

Here $r$ is whether the target (i.e., reward) is observed, $\alpha$ is the learning rate, and $\gamma$ is the discount factor. The term $\max_{a'} Q(s',a',c)$ represents the maximum Q-value for the next state $s'$, considering all possible actions $a'$. After updating the Q-table, the agent takes the next step to a new state $s'$ following an $\varepsilon$-greedy strategy. Specifically, the agent acts to maximize the expected Q-value 50% of the times, and explores randomly the remaining 50% of the times (i.e., $\varepsilon = 0.5$).

Importantly, the reward $r$ is received based on the following conditions: In the Skewed condition, $r = 1$ when the agent transitions to the predetermined target state, and $r = 0$ otherwise. In the Uniform condition, $r = 1$ only if the agent visits the cue state $c$ before transitioning to the target state. If this condition is not met, $r = 0$. Hence, visiting the cue functions as a flag, determining whether the subsequent reward (i.e. observing the target) will be obtained when visiting the target state. This aspect makes the reinforcement-learning model non-Markovian, as future steps depend on the past (i.e., whether the cue state was visited).

Q-values are updated after each step based on the reward received and the expected future rewards (which are a function of the Q-values of the next state). The updating formula inherently accounts for the delayed nature of the reward because it propagates the value of future rewards back to earlier states. In the uniform condition, the agent has no initial knowledge of the importance of the cue. However, through exploration, it occasionally visits the cue and then the target, receiving a reward. The Q-learning algorithm updates the Q-values to reflect this: the value of actions leading to the cue (when followed by actions leading to the target) increases. Over time, as the agent experiences more such successful sequences, the Q-values for actions leading to the cue state increase further, reflecting its importance in obtaining future rewards.

To determine the values of $\alpha$ and $\gamma$ that supported the most efficient search, we performed a grid-search algorithm across values of $\alpha$ (in the range from 0 to 1) and $\gamma$ (in the range from 0.5 to 1). Specifically, we trained each model over 300 episodes. For each episode, the state $s$, action $a$, and cue $c$ are initialized randomly from their respective domains. After training, the Q-values were fixed, such that no additional updating was possible, and we tested the model's ability to identify the target on 1000 additional episodes. We evaluated the model performance in terms of number of steps required to identify the target, where a lower number of steps indicates the ability to successfully identify the target in a shorter period of time. The most efficient search was achieved by the model with parameter values $\alpha = 0.12$ and $\gamma = .83$, with a mean number of 6.7 steps taken to find the target, which was 3.4 steps lesser than the average performance (10.1 steps).

We extracted the states visited by the most efficient model for both the Skewed and Uniform condition over the training episodes. These sequences of states indicate the model's exploration patterns, which are then compared to the participants' exploration patterns via Markov models. We included all episodes starting from the initial learning phase (in which the model is still unaware of the environmental structure) until the end of training (when the model has learned to navigate the environment successfully). This allowed us to focus on the episodes in which learning was occurring, making the results more comparable to data from humans, who also had to learn how to navigate the dark environments.

**Markov models.** The Markov model is a probabilistic model that handles sequential data by assuming that each observation is dependent solely on the current state of a discrete variable that evolves over time (i.e., as a Markov chain[41]). The Markov model is characterized by the number of states and transition probabilities, which determine the likelihood of transitioning from one state to another. Markov models were fitted on the sequences of eye movements to the AOIs on the screen. Specifically, four sequences of AOIs were obtained from the raw data, one for each age group (toddlers and adults) and condition (Skewed vs Uniform; i.e., 2×2 design). We ran each Markov model with 30 different initializations and number of iterations using the '*Multi-nomialHMM*' function of the *hmmlearn* Python package, and fixing the emission matrix to predetermined values, such that each of the 8 locations on the screen corresponded to one state of the Markov

model. This returned 30 different estimations of transitional probabilities for each of the four datasets. For each of these estimations, we computed their entropy according to the formula:

$$H(P) = - \sum_{i,j \in V} P_{i,j} \log(P_{i,j}) \qquad (2)$$

where the sum runs over all pairs $i,j$ of vertices $V$ between the states of the transition probability matrix $P$.

We used the entropy estimates as dependent variable in a linear model with age group, condition, and their interaction as predictors, and we compared the model to a null model that did not include any predictor. We reported results from the full model (AIC = 127), as it performed better than then null model (AIC = 278).

To compute the entropy of the most efficient search for the Skewed and Uniform condition, we used the sequence of states visited by the reinforcement-learning model as input for the Markov models. Again, the complexity of the most efficient search was computed by taking the entropy of the transitional probability matrix of the model's exploration patterns, separately for Skewed and Uniform condition. The complexity of random search was simply computed by taking the entropy of a transitional probability matrix in which all transitions were equally likely (i.e., 12.5%).

### Reporting summary
Further information on research design is available in the Nature Portfolio Reporting Summary linked to this article.

## Data availability
Data can be found on OSF: https://osf.io/rfx5u/[42].

## Code availability
Code for the task implementation, statistical analyses, and computational modeling can be found on OSF: https://osf.io/rfx5u/.

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

## Acknowledgements

We thank Andreas Domberg for his early contributions to the study design. Donders Centre for Cognition internal grant to S.H. and R.B.M. ("Here's looking at you, kid." A model-based approach to interindividual differences in infants' looking behavior and their relationship with cognitive performance and IQ; award/start date: 15 March 2018); Wellcome Trust center grant for R.B.M. ("Wellcome Centre for Integrative Neuroimaging"; award/start date: 30 October 2016; serial number: 203139/Z/16/Z); EPA Cephalosporin Fund and Biotechnology and Biological Sciences Research Council (BB/N019814/1) grants to R.B.M.

## Author contributions

F.P.: conceptualization, methodology, software, formal analysis, writing - original draft, and visualization. Y.L.: investigation, data curation, and writing - review & editing. P.N.: software, investigation, data curation, and writing - original draft. R.B.M.: conceptualization and writing - review & editing. S.H.: conceptualization and writing - review & editing. A.R.: conceptualization, resources, writing - review & editing, and supervision.

## Funding

## Competing interests

The authors declare no competing interests.
