## [Peer Review File · Nature Communications]

Toddlers strategically adapt their information searchREVIEWER COMMENTS

Reviewer #1 (Remarks to the Author):

The authors present an experiment which compares human behavior in two different search scenarios: in one, a hidden target may be in one of four different locations, indicated by visual cues which appear below the (hidden) target and deterministically predict the target's location. In the other scenario, the positions of the cue and target were always the same, so that an observer had no uncertainty as to their location. They compared the behavior of toddlers of different ages as well as adults across these conditions using eye-tracking measures. The primary finding is that both groups adapt their visual search to the condition: when the location of the cue and target (which are yoked) are randomized, observers look in more locations and make more eye movements while searching for the cue, and hence subsequently the target. They also compared human observers' behavior with predictions from a computational model.

This is an important topic, and my own research has asked related questions. However, my overall assessment of this work is largely negative. While I have no complaints about the study design (at least insofar as its validity), I don't think that the finding as presented adds much to our knowledge of cognitive development, either intuitively or with respect to previous literature. It may have merit as a methodological advance - I know little about eyetracking for visual search, either in toddlers or adults. But I think that few people with exposure to the developmental literature, or even laypeople with exposure to young children, would fail to predict results in line with the data. With an elaborated motivation and theoretical discussion, this could be fine, but as currently presented, in my opinion neither the motivation nor the results are sufficiently interesting to merit publication.

The theoretical motivation for the paper is the idea of active learning and exploration -- the idea that children's information-seeking choices are sensitive to, and respond sensibly to, the data that they receive and their own present epistemic state and needs. The manuscript does not discuss how the experimental task engages this theoretical question, but to me the interpretation of children's behavior as evidence of active learning is questionable. One might gloss the results as "when children know where to expect a (cue to a) target, they prefer to search in that location; when they don't know, they search more broadly until they find it". So while children do adapt their search strategy to incorporate acquired information about what to expect in future trials, I don't think that it demonstrates more sophisticated adaptive search than is seen in e.g. many standard visual search tasks. I suspect that a key point here is the use of the cue to direct the observer to the target, but I didn't understand the choice here and I think it could use more (some?) discussion.

The authors further include a computational model based on a Hidden Markov Model (although because the emission matrix is the identity matrix, it is equivalent to a standard Markov model - this should be clarified because I spent some time confused. In particular the emission matrix should be discussed). The model was used both for data analysis and is used to attempt to quantify the quality of participants' search behavior with respect to an "optimal" strategy (scare quotes to be explained). Without getting into the details, they compute a measure of the (inferred) diversity of participants' search strategies and compare it with the "optimal" strategy. This is a fine, if quite coarse, measure (entropy is a scalar measure of the diversity of a probability distribution, so does not evaluate the quality of any particular search sequence). The real issue here is with the derivation/specification of the "optimal" strategy. In computational neuroscience and cognitive science, one derives an optimal observer for a task via a statistical task specification along with a loss function which specifies the goal of the observer. In this manuscript, however, the authors analyze a particular search strategy, which they term optimal, without such an analysis. And it's not obvious to me that this strategy is optimal for this task. The language should be changed -- perhaps something like "heuristic strategy" could work -- or the authors should clarify how they derived their strategy according to some optimality principle. The model is also unusual in that it doesn't assess the overall quality of participant strategies, only their diversity -- while this is perhaps correlated with quality, it's not the same thing.

In conclusion, I think the manuscript (and the model) needs substantial work before it is ready for publication in a high profile journal. The task and results, while motivated by interesting and important questions, are not by themselves sufficiently interesting to sustain the paper, and I think it needs a rewrite with more theoretical motivation and more explanation of why the results are interesting and informative. If the model is intended to be a significant part of the story, it needs substantial work as well; if it is intended to be an ideal observer analysis in that tradition, as advertised by the "optimal" terminology, the authors need to demonstrate that it has that property. These are major revisions.

Notes:

- the PDF I have has a stray \leq sign on lines 154 and 155
- line 173: "waived and smiled" should be waved
- line 260: "estimation-maximization" should be "expectation-maximization"
- line 260: The expectation-maximization algorithm isn't gradient-based
- line 261: "optimal" should be "optima"
- Figure 1: the description of the task was difficult for me to follow.
- Figure 3: The emission matrix should be mentioned and discussed. In particular, the unusual use of an identity emission matrix meant that I had to carefully read the methods to understand the model. If the authors want to use the same model, they should simply refer to it as a Markov model, and defer the details (for example if they used a HMM software package to fit it) to the methods.

Reviewer #2 (Remarks to the Author):

This ms presents solid empirical and methodological advances that will lead to a better understanding of infants' cognitive abilities and their learning processes. The authors show that infants adapt search strategies to the informational structure of a situation, and, as of 2 years, that they have a close to optimal strategy. They do this with a novel paradigm in which infants control what they choose to see via a 'virtual torchlight'.

The experimental procedure is very, very clever. The analyses very well run. The results are quite clear. I am definitely in favor of its publication, it will attract interest and sparkle novel research.

I think some parts of the ms. still need work, though, mostly in three areas. First, a better connection with existing literature, and a more balance treatment of some of the quoted literature, should be tried. Then, some discussions of the meaning of the reported results would make it richer. Finally, some figures can be clarified a bit.

Below, I mention some of the points that could lead to a better revision.

-- Literature: there are several streams of evidence that go in the direction of the work presented here, but are not given the due consideration. Let me mention an example, but there could be others. Gweon et al (Gweon & Schulz, 2011) showed that infants younger than those tested here (14 mos) can decide the relevance of a source of information -- e.g., looking for help when they think they don't know how a device works, as opposed to looking for another device because they know how it works but it may be broken, or for another informant because one informant is not reliable, on the bases of a series of experiences. While not adaptive in the sense reported here, to me, the mental processes involved in these decisions have more than a family resemblance to those needed for toddlers to

succeed in the task presented in the ms.: in this case, infants also have to judge the reliability of a cue to get to a conclusion, and have to set up a correct strategy in the search for the best source of information. Comparing these kinds of data against the results presented in the ms would result in a better specification of their relevance. More specifically, while I praise the authors for their work, many questions are simply not tackled: what is **exactly** novel in their results? The fact that infants update their search strategy keeping track of the information presented across trials? Or the fact that they **have** a rational strategy to begin with? And if so, why <24 mos do not show a differential search strategy between the two conditions? How come they could be so good as deciding which source of information is more reliable in the Gweon & Schulz's case, and yet fail optimal search for information in the current MS? There are other results that could be relevant to better clarify what the contribution of this paper is exactly, and the authors can better review literature about young infants' anticipation/exploration/decision abilities and propose their story.

-- Between literature and theory: If I think of the reported results, in which infants dynamically adapt visual search to the nature of a cue (certain vs. equally likely), and I think about the failures that even 3 yr olds seem to undergo in apparently simpler cases involving essentially the same comparison (a comparison between a certain and a possible outcome; Mody & Carey, 2016; Leahy, Huemer, Steele, Alderete, & Carey, 2022), I am puzzled. I notice that this literature is absent from the paper. Could the authors say what their result mean wrt to these failures (more of this later)? What is the essential aspect of their design that makes 24 mos successful, and yet 3 yr olds fail simple decisions which conceptually involve the same elements? Is it the presence of frequency information in the familiarization (as opposed to the sheer conceptualization of the possibilities)? Is it the fact that their task does not involve any explicit choice or no action planning (besides eye movements)? In short, I perceive a conflict between the behavior the authors document here and the literature on early decision making. What do the author have to say about it?

-- Theory: The paper presents evidence (l. 124 ff) that toddlers adapt their oculomotor behavior to the information that can be gathered in a sequence of trials. A similar point has been made for how much younger infants anticipate future deterministic or probable outcomes (Teglas (Teglas & Bonatti, 2016). However, 12 month olds seem to be doing something partially in contrast with the current results: they do not anticipate deterministic outcomes, but only uncertain outcomes. In the current paper, instead, they seem to firmly make the distinction as of 24 months and anticipate accordingly. Teglas & Bonatti suggested that it was a question of information gain: infants gather no novel information in anticipating a deterministic outcome, whereas there is something at stake (they gain information) in anticipating an uncertain situation. So it looks like there is a shift in strategy between 12 and 24 months. Can the authors comment about it? Again, this is only an example; the point is to strive to connect the current results with what we know about early cognition in germane domains.

-- Theory: Just as I feel that the authors should make an effort in explaining success in their task at 24 months, in the light of evidence to the contrary coming from decision tasks, so they should try to consider the causes that lead infants to fail before 24 months. They say very little on this. One of the parameters entirely absent in their discussion is the number of alternatives presented. The design of the uniform condition dances along the limits of infants' ability to represent different numerical values or, if one believe infants can represent them, different potential alternatives. There is evidence that when the number of objects (Feigenson & Carey, 2005; Feigenson, Dehaene, & Spelke, 2004), or of possibilities (Teglas, Ibanez-Lillo, Costa, & Bonatti, 2014), exceeds 3 infants fail tasks that require to represent them. How do the author think that this dimension may influence the results in young toddlers? That is, is it possible that these infants cannot adopt different strategies in the skewed and uniform conditions, not because they are not sensitive to the difference between them per se, but simply because they can't represent four locations as distinct possibilities, and hence get confused? A good discussion of this interpretation could enrich the value of the paper, bridging literatures which have not been integrated.

For example, the authors could say that toddlers younger than 24 months simply do not get the

distinction between a fully reliable cue (deterministic; "The chest is in B") and an (equally probable) multiplicity of four possibilities ("The chest could be either in A or in B or in C or in D") because four are too many; the prediction here would be that a task with less locations (e.g. 3) would be feasible also in younger toddlers. Alternatively, they could argue, as it has been proposed by Carey and collaborators, that at this age infants simply have no notion of alternatives, and hence essentially treat the uniform condition exactly as the skewed condition, visiting one single of the possible locations of the chest because that's all they can represent (Mody & Carey, 2016; Leahy et al., 2022). Alternatively (but less likely), they could argue that young infants cannot adopt an optimal search strategy because they cannot correctly update the frequency distribution of the chest appearance, and hence don't get where to search initially. Or, perhaps they can say that the methodology implemented in the experiment may confuse the younger infants; perhaps they can't get how the spotlight works exactly. Indeed, what evidence shows that <24-mos understand the control part of the task?

Whichever their proposal, the authors should say something as to why young infants fail their task, considering the evidence for rational behavior gathered by many other authors at even younger ages.

-- Figures: Fig 1a is very cryptic to me. Besides presenting the material, I don't find it useful to understand the trial structure. I don't understand whether the grey squares correspond to real grey levels superimposed onto the stimuli, or they are just a qualitative indication that the background changes. The authors should clarify. I don't understand the numbers marked by the animals in fig 1A, as they are never mentioned later. Fig 1.B: what is it? an example of a specific trial? What is the lighter area? the infant-controlled torch? And why does it appear where it does? Please clarify. In Fig 2, I think it would be useful to add the individual data points (means per subject) in panels A and B.

Other points:

-- Sample size of the <24 and >24 groups: I understand the <24 mos and >24 mos is a split made post-hoc, given the results of the difference between conditions. However, I do not find the number of ss in each subgroup. These data should be reported.

--l. 25: maybe starting the paper with a quote of a (preprint) related to AI, which is not a topic ever talked about in the paper, is not the most felicitous choice. Furthermore, at least from the way I understand it, it is not clear on the basis of which part of the (quoted) Forestier et al's paper it can be claimed that human exploration abilities are superior to the artificial agent. Explain, if relevant, or remove. Ditto for the abstract, where AI is mentioned again. What does AI have to do with the ms? Explain or remove.

-- l. 35 ff: This part is very problematic to me. I don't get how the a-not b results i.e., failures at 9 months, success after 12 months) have anything to do with the toddlers' search strategies studied in the ms., or for that matter, with cognitive abilities in general. There is a humungous literature related to this task and simply saying that infants fail it (without even quoting the context and the very well documented reasons for failures and successes) gives the impression that the authors are setting up a straw man. I have the same perception when I see that they report a case of young infants matching probabilities instead of maximizing outcomes in a visual search task, which, with all due respect, appeared in the proceedings of a Cognitive Science meetings, disregarding the considerable literature documenting their understand many probability concepts, the relations between sample and populations, and many other concepts that go well beyond probability matching strategies. I feel that the reference to a single, not particularly crucial paper (6 quotes in Google Scholar) and disregarding the full literature about probabilistic abilities, is not particularly convincing. Again, it seems like the authors are making some sort of rhetorical point rather than making a substantial argument. I would reconsider this full paragraph, which is going to raise many eyebrows.

-- l. 63: "Both toddlers and adults performed the task successfully, correctly identifying the location of the hidden animal in both conditions". Where is the statistical/graphical support for this statement? If

it is so important as to open the result section, it should be reported in the main text and not in the supp mat. The statement is repeated in the discussion (l.131: "We found that active information search is already successful at 18 months of age, as indicated by toddlers' ability to find the target by exploiting the informative cue."

-- Supp mat: it is missing for what I can see, and so I could not examine it.

-- l.124: ""[...] we demonstrated that this ability ... emerges earlier ... than previously demonstrated." Maybe there is a better phrasing.

-- l. 134: "Although previous studies identified rudimentary forms of active learning from early infancy": this is a very generic statement, from which I personally cannot gather what the authors have in mind. At least, they should provide some direct references to some scientific articles (as opposed to a general encyclopedia entry), so that the reader understands what the authors have in mind, what these demonstrations are and why they are rudimentary.

-- l. 139: what is ref 23? published? unpublished? One cannot say from the quoted ref.

-- (l. 62 ff): Fig 1 is Fig 2, clearly.

-- l 154-5 : Typos. You mean a sample of 59 toddlers and of 28 adults?

-- in the osf storage I found the data but not the analysis files, so I could not check the analyses.

-- (l. 315) the number of the quotes 7 and 10 are missing in my version of the ms (pdf). Also, in the pdf version Figure 3 is severely degraded, half-eaten up, probably because of the pdf conversion on the site. I had to go to the original version to see it.

====

Quoted Papers

Feigenson, L., & Carey, S. (2005). On the limits of infants' quantification of small object arrays. *Cognition*, 97(3), 295-313.

Feigenson, L., Dehaene, S., & Spelke, E. (2004). Core systems of number. *Trends in Cognitive Sciences*, 8(7), 307-314.

Gweon, H., & Schulz, L. (2011). 16-month-olds rationally infer causes of failed actions. *Science*, 332(6037), 1524.

Leahy, B., Huemer, M., Steele, M., Alderete, S., & Carey, S. (2022). Minimal representations of possibility at age 3. *Proceedings of the National Academy of Sciences of the United States of America*, 119(52), e2207499119.

Mody, S., & Carey, S. (2016). The emergence of reasoning by the disjunctive syllogism in early childhood. *Cognition*, 154, 40-48.

Téglás, E., & Bonatti, L. L. (2016). Infants anticipate probabilistic but not deterministic outcomes. *Cognition*, 157, 227-236.

Teglas, E., Ibanez-Lillo, A., Costa, A., & Bonatti, L. L. (2014). Numerical representations and intuitions of probabilities at 12 months. *Dev Sci*.

Luca Lorenzo Bonatti.

Reviewer #3 (Remarks to the Author):

This is a potentially very interesting paper, showing that toddlers, from age at least aged 24 months onward alter their visual exploration behavior based on the nature of a search task.

I found the paper to be well written and thorough in its coverage of past literature, and I greatly appreciated the comparisons to optimal search generated by HMMs.

There was one aspect of the paper that I had a hard time following that I think matters critically for the nature of toddlers' performance on the task and for the novelty of these findings. Save the darkening screen, I had a hard discerning whether the training trials were different from the test trials, and critically, whether training differed across conditions. My central concern here is that if training was different across the conditions (i.e., the animal always appeared in one column for the uniform condition, but appeared in multiple columns across different trials in the skewed condition), but identical to the structure of the task on test, the findings may simply reflect a conditioning process (i.e., toddlers are rewarded during training for scanning broadly in the uniform than skewed condition; these behaviors then generalize to test trials). Essentially, more information about the similarity/dissimilarity between training and test trials is necessary to determine the novelty and ultimate contribution of the findings.

Reviewer #1 (Remarks to the Author):

The authors present an experiment which compares human behavior in two different search scenarios: in one, a hidden target may be in one of four different locations, indicated by visual cues which appear below the (hidden) target and deterministically predict the target's location. In the other scenario, the positions of the cue and target were always the same, so that an observer had no uncertainty as to their location. They compared the behavior of toddlers of different ages as well as adults across these conditions using eye-tracking measures. The primary finding is that both groups adapt their visual search to the condition: when the location of the cue and target (which are yoked) are randomized, observers look in more locations and make more eye movements while searching for the cue, and hence subsequently the target. They also compared human observers' behavior with predictions from a computational model.

1. This is an important topic, and my own research has asked related questions. However, my overall assessment of this work is largely negative. While I have no complaints about the study design (at least insofar as its validity), I don't think that the finding as presented adds much to our knowledge of cognitive development, either intuitively or with respect to previous literature. It may have merit as a methodological advance - I know little about eyetracking for visual search, either in toddlers or adults. But I think that few people with exposure to the developmental literature, or even laypeople with exposure to young children, would fail to predict results in line with the data. With an elaborated motivation and theoretical discussion, this could be fine, but as currently presented, in my opinion neither the motivation nor the results are sufficiently interesting to merit publication.

The theoretical motivation for the paper is the idea of active learning and exploration -- the idea that children's information-seeking choices are sensitive to, and respond sensibly to, the data that they receive and their own present epistemic state and needs. The manuscript does not discuss how the experimental task engages this theoretical question, but to me the interpretation of children's behavior as evidence of active learning is questionable. One might gloss the results as "when children know where to expect a (cue to a) target, they prefer to search in that location; when they don't know, they search more broadly until they find it". So while children do adapt their search strategy to incorporate acquired information about what to expect in future trials, I don't think that it demonstrates more sophisticated adaptive search than is seen in e.g. many standard visual search tasks. I suspect that a key point here is the use of the cue to direct the observer to the target, but I didn't understand the choice here and I think it could use more (some?) discussion.

- 1.1. We thank Reviewer 1 (R1) for agreeing that the topic and research questions we investigate in this paper are important and timely. We are glad that R1 finds our study design sound and has no criticisms concerning its validity. We are also thankful to R1 for giving us the opportunity to further expand on the theoretical background of our study, and for giving us constructive suggestions on how to do so. To integrate them into the paper, we have now added two paragraphs to the introduction.

First, we decided to state more clearly the research questions of the current work. In particular, we added the following paragraph to the manuscript (p.1, line 40 – p. 2, line 3):

“Evidence of the ability to actively devise flexible information-seeking strategies is lacking in children below the age of 3. In particular, it is an open question whether infants realize that a piece of information can be instrumental to constrain the search of a target, thus allowing to achieve a certain goal more efficiently, whether they search for information independently and unprompted, and whether they change their search strategy depending on the specific

probabilistic structure of the environment.”

These issues have never been explored. In this sense, we provide the first evidence on the emergence of adaptive information seeking in infancy. Specifically, we show that already at 18 months of age, infants can realize that a certain piece of information can help them finding the target (point 1). Indeed, they already perform above chance at this age. However, only from 24 months of age onwards, they start to strategically search for the informative cue (point 2) and to tailor their information-seeking strategies to the characteristics of the environment they have been presented with (point 3).

1.2. Second, we put our research question in direct comparison with what we already know about the development of information-seeking abilities, explaining why this study is crucial to understanding the developmental emergence of *adaptive* information-seeking (p. 2, lines 8-14):

“During the second year of life, toddlers are more likely to solicit information from a knowledgeable adult compared to an ignorant or unreliable one^{18,19}, or when presented with more cognitively demanding tasks²⁰. After failing to activate a toy, 16-month-old infants seek for help if they have reason to believe that the failure is due to their own inability, but they explore different objects if they have reason to believe that the object is malfunctioning²¹. These studies, though relevant, do not provide evidence as to whether infants’ information-seeking strategies are adaptive, in the sense that they are tailored to the characteristics of the environment they are presented with.”

1.3. Finally, we offer an overview of the ecological active learning perspective, which has previously investigated the adaptiveness of learning and exploration strategies in children and adults (p. 2, lines 15-26):

“This adaptive competence has been investigated only in older children and adults, relying on tasks that require participants to tailor their search strategies to the changing characteristics of the environment—for example, to flexibly decide whether to engage in hypothesis-probing or constraint-seeking questions⁷. Hypothesis-probing questions target specific individual hypotheses (e.g., “Is it the penguin?” when trying to find out what animal can endure the lowest temperatures). In contrast, constraint-seeking questions aim to narrow down the range of hypotheses under consideration by testing higher-level characteristics shared by multiple hypotheses (e.g., “Does it have wings?”). School-aged children are more likely to rely on hypothesis-probing questions when there is a most likely hypothesis they can target (Skewed hypothesis space), but on constraint-seeking questions when all the hypotheses under consideration are equally likely (Uniform hypothesis space)⁵, and 3- and 4-year-old children show a similar pattern of results in a nonverbal version of the same paradigm²², indicating that they are indeed able to adapt their exploration strategies to the statistical structure of a task.”

1.4. The summary of the results offered by the reviewer (i.e., “when children know where to expect a (cue to a) target, they prefer to search in that location; when they don't know, they search more broadly until they find it”) is correct, but incomplete. We do not just find that infants change the *amount of overall scanning behavior* depending on the condition, that is, on the level of uncertainty. In fact, by design, scanning the target locations (i.e., the upper part of the screen) in the Uniform condition would lead to failure in finding the target. Conversely, we observe an increase in visual scanning in the Uniform condition that is directed to the cue area, demonstrating that infants behave strategically, looking for information when they need it.

- 1.5. An additional novel aspect is the gaze-contingent paradigm. We developed the “torchlight” paradigm to allow toddlers to freely explore while eliminating the need for explicit choices or overt actions. This new paradigm greatly simplifies the task, and it allows us to test toddler’s information-seeking while tapping as little as possible into other cognitive abilities. We now stress this aspect more in the discussion, p. 5, lines 9-17 (also following the suggestion from R2, see point 2.1 in the response to R2):

“Previous research indicates that children under the age of 3 struggle to effectively narrow down their search for a hidden reward among multiple options^{28,29}. For instance, in a recent study²⁹, toddlers were presented with four cups (A, B, C, and D) arranged in two pairs (A and B versus C and D). One cup in each pair contained a hidden sticker, but the specific cup was unknown. When one cup was revealed to be empty (e.g., cup B), toddlers should have realized that finding a sticker in cup A was certain, while finding one in cup C or D was only possible. However, their behavior did not demonstrate a preference for cup A. Although similar in structure to our task, these studies differed in two important ways. First, success in previous tasks required logical abilities that relied on abstract, combinatorial thought, while the current task required probabilistic reasoning about statistical regularities—an ability that emerges earlier in infancy³⁰; Second, the current task does not involve any explicit choice or action planning besides eye movements, while previous studies required both. These simplifications allowed us to capture the early emergence of adaptive information seeking, tapping as little as possible into other competences.”

Thus, we present a new methodological approach in the developmental literature on information-seeking, which allowed us to answer important questions on the development of adaptive information-seeking abilities. This is a methodological advance that will allow also other researchers to easily test information-seeking in toddlerhood and possibly infancy, which has been nearly impossible so far.

2. The authors further include a computational model based on a Hidden Markov Model (although because the emission matrix is the identity matrix, it is equivalent to a standard Markov model - this should be clarified because I spent some time confused. In particular the emission matrix should be discussed). The model was used both for data analysis and is used to attempt to quantify the quality of participants' search behavior with respect to an "optimal" strategy (scare quotes to be explained). Without getting into the details, they compute a measure of the (inferred) diversity of participants' search strategies and compare it with the "optimal" strategy. This is a fine, if quite coarse, measure (entropy is a scalar measure of the diversity of a probability distribution, so does not evaluate the quality of any particular search sequence). The real issue here is with the derivation/specification of the "optimal" strategy. In computational neuroscience and cognitive science, one derives an optimal observer for a task via a statistical task specification along with a loss function which specifies the goal of the observer. In this manuscript, however, the authors analyze a particular search strategy, which they term optimal, without such an analysis. And it's not obvious to me that this strategy *is* optimal for this task. The language should be changed -- perhaps something like "heuristic strategy" could work -- or the authors should clarify how they derived their strategy according to some optimality principle. The model is also unusual in that it doesn't assess the overall quality of participant strategies, only their diversity -- while this is perhaps correlated with quality, it's not the same thing.

- 2.1. We thank R1 for giving us the opportunity to clarify these points. First of all, thanks to R1’s comment we have now realized that, by fixing the emission matrix to predetermined values, we remove the hidden states and make all the information directly observable. In this sense, we agree that it is correct to refer to the model as a Markov model, where the system's behavior is governed by the transitions between directly observable states, instead of Hidden Markov

model. We have implemented this change throughout the paper. We also removed the emission matrix from Figure 3, and changed the explanation of the model in the methods section (p. 9, lines 4-7):

“The Markov model is a probabilistic model that handles sequential data by assuming that each observation is dependent solely on the current state of a discrete variable that evolves over time (i.e., as a Markov chain⁴¹). The Markov model is characterized by the number of states and transition probabilities, which determine the likelihood of transitioning from one state to another.”

We would like to stress that, although we have now corrected the model label, all other aspects of the model remain intact, so this change has no impact on the results and the conclusions.

2.2. We agree with R1 that our simulated *optimal* strategies are different from what is usually referred to as optimal observer and that “efficient” is likely a more accurate characterization: In the Skewed condition, the most *efficient* search consists in moving to the high-likelihood location directly, whereas in the Uniform condition, the most *efficient* search consists in scanning for the cue and once found, move towards the target location. The algorithms we implemented carry out these two processes, and behave in line with our expectations about efficient search. Hence, to avoid any confusion with ideal observer models, we changed the label “optimal” to “most efficient” throughout the paper.

2.3. Finally, we agree that the measure of entropy that we compute over the transition matrices of the Markov model allows us to measure the *diversity* (or complexity) of the probability distribution, not its quality. In the Uniform condition, we show that the transition matrices are more “diverse” (or “complex”) than in the Skewed condition. The measure of entropy allows us to support this conclusion. However, we can draw conclusions about differences in the *quality* of information-seeking patterns from the other analyses in the paper. In particular, the behavioral analyses of scanning and anticipatory looks show that participants engaged in information-seeking strategies that are qualitatively different across conditions. The Markov model offers additional evidence in this direction. Indeed, a direct comparison of the transitional probabilities of adults and toddlers (see Figure 3) allows us to show that toddlers are more likely to transition to a state in the bottom half of the screen, hence indicating that they are more likely to seek information than adults. We appreciate that we did not highlight this result well enough, and therefore we have now added an additional Figure illustrating this point (new Figure 3D), which we also report it in the discussion (p. 5, lines 39-41):

“The exploration patterns detected by the Markov models indicate that 24-to-36-month-old toddlers’ exploration was driven to the informative cue locations more so than adults’ exploration, demonstrating that toddlers displayed a greater tendency towards gaining information”

3. Notes:

- the PDF I have has a stray <= sign on lines 154 and 155
- line 173: "waived and smiled" should be waved
- line 260: "estimation-maximization" should be "expectation-maximization"
- line 260: The expectation-maximization algorithm isn't gradient-based
- line 261: "optimal" should be "optima"

We have implemented all changes as suggested by R1. We have also added the package and

function that we used to fit the model (see below).

- Figure 1: the description of the task was difficult for me to follow.

We have improved the description of the task by describing each element in the figure, while reducing the overall length of the caption.

- Figure 3: The emission matrix should be mentioned and discussed. In particular, the unusual use of an identity emission matrix meant that I had to carefully read the methods to understand the model. If the authors want to use the same model, they should simply refer to it as a Markov model, and defer the details (for example if they used a HMM software package to fit it) to the methods.

Since in Markov models the emission matrix corresponds to the identity matrix, we have now removed the emission matrix for the figure. We now also refer to the models as Markov models (instead of HMM), and specify the package and function we used to fit the model (p.9, lines 10-13):

"We ran each Markov model with 30 different initializations and number of iterations using the 'MultinomialHMM' function of the hmmlearn python package, fixing the emission matrix to predetermined values such that each of the 8 locations on the screen corresponded to one state of the Markov model."

Reviewer #2 (Remarks to the Author):

This ms presents solid empirical and methodological advances that will lead to a better understanding of infants' cognitive abilities and their learning processes. The authors show that infants adapt search strategies to the informational structure of a situation, and, as of 2 years, that they have a close to optimal strategy. They do this with a novel paradigm in which infants control what they choose to see via a 'virtual torchlight'.

The experimental procedure is very, very clever. The analyses very well run. The results are quite clear. I am definitely in favor of its publication, it will attract interest and sparkle novel research.

I think some parts of the ms. still need work, though, mostly in three areas. First, a better connection with existing literature, and a more balance treatment of some of the quoted literature, should be tried. Then, some discussions of the meaning of the reported results would make it richer. Finally, some figures can be clarified a bit.

Below, I mention some of the points that could lead to a better revision.

1. -- Literature: there are several streams of evidence that go in the direction of the work presented here, but are not given the due consideration. Let me mention an example, but there could be others. Gweon et al (Gweon & Schulz, 2011) showed that infants younger than those tested here (14 mos) can decide the relevance of a source of information -- e.g., looking for help when they think they don't know how a device works, as opposed to looking for another device because they know how it works but it may be broken, or for another informant because one informant is not reliable, on the bases of a series of experiences. While not adaptive in the sense reported here, to me, the mental processes involved in these decisions have more than a family

resemblance to those needed for toddlers to succeed in the task presented in the ms.: in this case, infants also have to judge the reliability of a cue to get to a conclusion, and have to set up a correct strategy in the search for the best source of information. Comparing these kinds of data against the results presented in the ms would result in a better specification of their relevance. More specifically, while I praise the authors for their work, many questions are simply not tackled: what is **exactly** novel in their results? The fact that infants update their search strategy keeping track of the information presented across trials? Or the fact that they **have** a rational strategy to begin with? And if so, why <24 mos do not show a differential search strategy between the two conditions? How come they could be so good as deciding which source of information is more reliable in the Gweon & Schulz's case, and yet fail optimal search for information in the current MS? There are other results that could be relevant to better clarify what the contribution of this paper is exactly, and the authors can better review literature about young infants' anticipation/exploration/decision abilities and propose their story.

- 1.1. We thank the reviewer for giving us the opportunity to better describe the novelty of the presented paradigm. We believe that our study design and paradigm differ in important ways from the existing literature on infants' exploratory behavior, and we now address this issue more explicitly. First, we state our research questions more explicitly, and address how they differ from existing work (p. 1, line 40 – p. 2, line 14):

“Evidence of the ability to actively devise flexible information-seeking strategies is lacking in children below the age of 3. In particular, it is an open question whether infants realize that a piece of information can be instrumental to constrain the search of a target, thus allowing to achieve a certain goal more efficiently, whether they search for information independently and unprompted, and whether they change their search strategy depending on the specific probabilistic structure of the environment. Previous work suggests that some of the fundamental skills that support adaptive information-seeking can be traced back to the first years of life. Already at 8 months of age, infants can learn complex probabilistic and hierarchical structures from a stream of incoming stimuli^{13,14}, and by the end of their first year, they engage in active exploration when new events violate their expectations^{15,16} or promise an information gain¹⁷. During the second year of life, toddlers are more likely to solicit information from a knowledgeable adult compared to an ignorant or unreliable one^{18,19}, or when presented with more cognitively demanding tasks²⁰. After failing to activate a toy, 16-month-old infants seek for help if they have reason to believe that the failure is due to their own inability, but they explore different objects if they have reason to believe that the object is malfunctioning²¹. These studies, though relevant, do not provide evidence as to whether infants' information-seeking strategies are adaptive, in the sense that they are tailored to the characteristics of the environment they are presented with.”

This also speaks to the differences between our study and the one by Gweon and Schulz (2011) mentioned by R2. In our study, infants need to devise actions to gather information (i.e., find the informative cue) with the goal to constrain their search for information (i.e., finding the location of the target animal). In Gweon and Schulz (2011), actions are devised to maximize their success to operate the toy, rather than to gather information by devising a specific information-seeking strategy.

- 1.2. Second, we connect the current work to previous research on older children and adults, with the additional goal of clarifying the theoretical framework (p. 2, lines 15-26):

“This adaptive competence has been investigated only in older children and adults, relying on tasks that require participants to tailor their search strategies to the changing characteristics of the environment—for example, to flexibly decide whether to engage in hypothesis-probing or

*constraint-seeking questions*⁷. *Hypothesis-probing questions target specific individual hypotheses (e.g., “Is it the penguin?” when trying to find out what animal can endure the lowest temperatures). In contrast, constraint-seeking questions aim to narrow down the range of hypotheses under consideration by testing higher-level characteristics shared by multiple hypotheses (e.g., “Does it have wings?”). School-aged children are more likely to rely on hypothesis-probing questions when there is a most likely hypothesis they can target (Skewed hypothesis space), but on constraint-seeking questions when all the hypotheses under consideration are equally likely (Uniform hypothesis space)*⁵, and 3- and 4-year-old children show a similar pattern of results in a nonverbal version of the same paradigm²², indicating that they are indeed able to adapt their exploration strategies to the statistical structure of a task.”

2. -- Between literature and theory: If I think of the reported results, in which infants dynamically adapt visual search to the nature of a cue (certain vs. equally likely), and I think about the failures that even 3 yr olds seem to undergo in apparently simpler cases involving essentially the same comparison (a comparison between a certain and a possible outcome; Mody & Carey, 2016; Leahy, Huemer, Steele, Alderete, & Carey, 2022), I am puzzled. I notice that this literature is absent from the paper. Could the authors say what their result mean wrt to these failures (more of this later)? What is the essential aspect of their design that makes 24 mos successful, and yet 3 yr olds fail simple decisions which conceptually involve the same elements? Is it the presence of frequency information in the familiarization (as opposed to the sheer conceptualization of the possibilities)? Is it the fact that their task does not involve any explicit choice or no action planning (besides eye movements)? In short, I perceive a conflict between the behavior the authors document here and the literature on early decision making. What do the author have to say about it?

- 2.1. We thank the reviewer for pointing out previous work on logical reasoning that relates to our study. We now added a new paragraph to the discussion, in which we address the potential reasons for children under 3 succeeding on our task, but failing in those tasks mentioned by R2 (p. 5, lines 9-21):

“Previous research indicates that children under the age of 3 struggle to effectively narrow down their search for a hidden reward among multiple options^{28,29}. For instance, in a recent study²⁹, toddlers were presented with four cups (A, B, C, and D) arranged in two pairs (A and B versus C and D). One cup in each pair contained a hidden sticker, but the specific cup was unknown. When one cup was revealed to be empty (e.g., cup B), toddlers should have realized that finding a sticker in cup A was certain, while finding one in cup C or D was only possible. However, their behavior did not demonstrate a preference for cup A. Although similar in structure to our task, these studies differed in two important ways. First, success in previous tasks required logical abilities that relied on abstract, combinatorial thought, while the current task required probabilistic reasoning about statistical regularities—an ability that emerges earlier in infancy³⁰; Second, the current task does not involve any explicit choice or action planning besides eye movements, while previous studies required both. These simplifications allowed us to capture the early emergence of adaptive information seeking, tapping as little as possible into other competences.”

This is related to our following response (point 3.1).

3. -- Theory: The paper presents evidence (l. 124 ff) that toddlers adapt their oculomotor behavior to the information that can be gathered in a sequence of trials. A similar point has been made for how much younger infants anticipate future deterministic or probable outcomes (Teglas (Teglas & Bonatti, 2016). However, 12 month olds seem to be doing something partially in contrast with the current results: they do not anticipate deterministic outcomes, but only

uncertain outcomes. In the current paper, instead, they seem to firmly make the distinction as of 24 months and anticipate accordingly. Teglas & Bonatti suggested that it was a question of information gain: infants gather no novel information in anticipating a deterministic outcome, whereas there is something at stake (they gain information) in anticipating an uncertain situation. So it looks like there is a shift in strategy between 12 and 24 months. Can the authors comment about it? Again, this is only an example; the point is to strive to connect the current results with what we know about early cognition in germane domains.

3.1. We believe that there is a crucial difference between our study and that by Teglas & Bonatti (2016). There, deterministic outcomes were obtained by posing a hard constraint (i.e., a wall with no openings). Conversely, our Skewed condition was obtained by accumulating statistical evidence (over 8 familiarization trials) indicating that one location is *more likely* than the others. In this sense, because there is always a chance that this statistical evidence may be violated, we cannot refer to the two conditions as deterministic vs. probabilistic. Hence, we expect that infants in our task always reasoned under probabilistic conditions, and this is why we find results consistent with Teglas & Bonatti (2016). We were inspired by that study when analyzing our results, and we now cite it accordingly (p. 3, line 25):

"Following previous work²⁶, we analyzed whether participants performed anticipatory looks (i.e., looks before the start of the search phase) onto the correct portion of the screen (i.e., column)."

Moreover, we agree that it is important to further clarify the distinction between "Skewed" and "deterministic." We therefore removed the potentially confusing statement that the Skewed condition was "a more deterministic environment" and instead now say that it was a "more predictable environments" (p. 3, lines 43-44).

4. -- Theory: Just as I feel that the authors should make an effort in explaining success in their task at 24 months, in the light of evidence to the contrary coming from decision tasks, so they should try to consider the causes that lead infants to fail before 24 months. They say very little on this. One of the parameters entirely absent in their discussion is the number of alternatives presented. The design of the uniform condition dances along the limits of infants' ability to represent different numerical values or, if one believe infants can represent them, different potential alternatives. There is evidence that when the number of objects (Feigenson & Carey, 2005; Feigenson, Dehaene, & Spelke, 2004), or of possibilities (Teglas, Ibanez-Lillo, Costa, & Bonatti, 2014), exceeds 3 infants fail tasks that require to represent them. How do the author think that this dimension may influence the results in young toddlers? That is, is it possible that these infants cannot adopt different strategies in the skewed and uniform conditions, not because they are not sensitive to the difference between them per se, but simply because they can't represent four locations as distinct possibilities, and hence get confused? A good discussion of this interpretation could enrich the value of the paper, bridging literatures which have not been integrated.

For example, the authors could say that toddlers younger than 24 months simply do not get the distinction between a fully reliable cue (deterministic; "The chest is in B") and an (equally probable) multiplicity of four possibilities ("The chest could be either in A or in B or in C or in D") because four are too many; the prediction here would be that a task with less locations (e.g. 3) would be feasible also in younger toddlers. Alternatively, they could argue, as it has been proposed by Carey and collaborators, that at this age infants simply have no notion of alternatives, and hence essentially treat the uniform condition exactly as the skewed condition, visiting one single of the possible locations of the chest because that's all they can represent (Mody & Carey, 2016; Leahy et al., 2022). Alternatively (but less likely), they could argue that young infants cannot adopt an optimal search strategy because they cannot correctly update the

frequency distribution of the chest appearance, and hence don't get where to search initially. Or, perhaps they can say that the methodology implemented in the experiment may confuse the younger infants; perhaps they can't get how the spotlight works exactly. Indeed, what evidence shows that <24-mos understand the control part of the task?

Whichever their proposal, the authors should say something as to why young infants fail their task, considering the evidence for rational behavior gathered by many other authors at even younger ages.

- 4.1. We thank the reviewer for proposing multiple explanations of what might underlie the failure in our task for children below 24 months. In the previous version of the manuscript, we briefly mentioned some of the reasons that may lead to failure in younger toddlers, but now we describe them in more detail (p. 5 lines 26-36):

“This developmental change might depend on multiple factors. First, recent findings show that infants possess the ability to exploit past experiences to discover better learning strategies^{32,33}. After 8-month-old infants learned that multiple environments shared the same probabilistic structure, they were able to rely on this knowledge to quickly identify informative stimuli in new environments³². In this sense, adaptive information-seeking competence might be gradually learned after failing with simpler, more rigid exploration strategies, such as random or hypothesis-probing exploration. Second, cognitive flexibility is still drastically developing across toddlerhood^{34,35}. A key challenge for younger children is resolving how to respond flexibly when a task presents conflicting information, such as conflicting rules or bivalent stimuli³⁵. These skills develop across the second and third year of life³⁶, and they might be at the foundation of toddlers’ emerging ability to flexibly switch information-seeking strategy based on the different probabilistic structure of each environment.”

- 4.2. The number of alternatives that infants can entertain is also an important aspect of our task. We overlaid each target and cue location with a white dot (see Fig. 1B) to ensure that, even when the screen was completely dark, infants would have a reminder of the task-relevant locations. This made the task less reliant on memory processes. Anecdotally, these 8 white dots were absent during piloting of the task, and we observed that infants had problems exploring. Hence, we think that the presence of these placeholders prevents the alternative explanation offered by the reviewer.

Regarding R2’s last point, we cannot directly exclude the possibility that younger toddlers may have not understood that they had control over the torchlight. However, several pieces of evidence point to the opposite direction. First, although younger toddlers did not adapt their search to the experimental conditions, they succeeded in finding the target in both conditions. This pattern of results suggests that when the infants’ gaze encountered the cue, they were then able to actively move the torch towards the target location. Hence, the key aspect that led to failure does not seem to be a lack of understanding of the gaze-contingent torch, but the inability to actively design an efficient search strategy to find the informative cue. Second, a previous study with 8-month-olds used a “image scratch” paradigm, in which an image could “scratch” a gray cover with their eyes to reveal the image underneath (Miyazaki et al., 2014). Their analyses showed that infants’ looking behavior during scratching was more similar to adults’ active and spontaneous looking in the same task, than to adults’ passive looking. This supports the possibility that gaze-contingent studies would work from early in infancy.

Overall, we think that our task design and our pattern of results allow us to discard alternative explanations of our findings, and rather point to the direction that the reason of younger toddler’s failure is due to limits in exploratory strategies or flexibility – that is, what we set out to test.

Reference:

Miyazaki, M., Takahashi, H., Rolf, M., Okada, H., & Omori, T. (2014). The image-scratch paradigm: a new paradigm for evaluating infants' motivated gaze control. *Scientific reports*, 4(1), 1-6.

5. -- Figures: Fig 1a is very cryptic to me. Besides presenting the material, I don't find it useful to understand the trial structure. I don't understand whether the grey squares correspond to real grey levels superimposed onto the stimuli, or they are just a qualitative indication that the background changes. The authors should clarify. I don't understand the numbers marked by the animals in fig 1A, as they are never mentioned later. Fig 1.B: what is it? an example of a specific trial? What is the lighter area? the infant-controlled torch? And why does it appear where it does? Please clarify. In Fig 2, I think it would be useful to add the individual data points (means per subject) in panels A and B.

- 5.1. We have now added an explanation of the numbers accompanying the animals and of the gray squares of Figure 1A:

"Figure 1. A. Participants were presented with two within-subject conditions, Skewed and Uniform, each consisting of two blocks (numbered 1 to 4) composed of training and test trials. Subsequent trials had an increasing level of darkness (a qualitative indication is reported by the squares on the right)."

- 5.2. Moreover, we have added individual data points (means per subject) in panel A of Figure 2, and reported this in the caption.

Figure 2. Scanning movements and anticipatory looks. A. Predictive estimates of the number of scanning movements across age show an interaction between age and condition. Dots indicate the mean number of trials for each infant.

Panel B represents the difference in between the slopes that are estimated from the beta coefficients that are reported in panel A. For this reason, we cannot overlay data to the figure.

Other points:

6. -- Sample size of the <24 and >24 groups: I understand the <24 mos and >24 mos is a split made post-hoc, given the results of the difference between conditions. However, I do not find the number of ss in each subgroup. These data should be reported.

We now report the number of participants in each group, as follows (p. 4, lines 8-9):

“Toddlers below 24 months of age (who did not show adaptive behavior from the behavioral analyses, N = 14), toddlers above 24 months of age (N = 46), and adults (N = 42)”

7. --l. 25: maybe starting the paper with a quote of a (preprint) related to AI, which is not a topic ever talked about in the paper, is not the most felicitous choice. Furthermore, at least from the way I understand it, it is not clear on the basis of which part of the (quoted) Forestier et al's paper it can be claimed that human exploration abilities are superior to the artificial agent. Explain, if relevant, or remove. Ditto for the abstract, where AI is mentioned again. What does AI have to do with the ms? Explain or remove.

We agree with R2 that artificial intelligence does not play a key role in our paper. We removed the reference to it in the abstract. We improved the citations in the first sentence of the introduction (see below). Although we agree that artificial intelligence is not a central aspect of our paper, we still think that the first sentence nicely sets up the stage for highlighting that this adaptive exploration ability is (still) unique and exclusive to humans.

New citations:

- Zador, A. M. (2019). A critique of pure learning and what artificial neural networks can learn from animal brains. *Nature communications*, 10(1), 3770. (<https://www.nature.com/articles/s41467-019-11786-6>)
 - Sinz, F. H., Pitkow, X., Reimer, J., Bethge, M., & Tolias, A. S. (2019). Engineering a less artificial intelligence. *Neuron*, 103(6), 967-979. (<https://www.sciencedirect.com/science/article/pii/S0896627319307408>)
 - Forestier, S., Portelas, R., Mollard, Y., & Oudeyer, P. Y. (2022). Intrinsically motivated goal exploration processes with automatic curriculum learning. *The Journal of Machine Learning Research*, 23(1), 6818-6858. (<https://dl.acm.org/doi/pdf/10.5555/3586589.3586741>)
8. -- l. 35 ff: This part is very problematic to me. I don't get how the a-not b results (i.e., failures at 9 months, success after 12 months) have anything to do with the toddlers' search strategies studied in the ms., or for that matter, with cognitive abilities in general. There is a humungous literature related to this task and simply saying that infants fail it (without even quoting the context and the very well documented reasons for failures and successes) gives the impression that the authors are setting up a straw man. I have the same perception when I see that they report a case of young infants matching probabilities instead of maximizing outcomes in a visual search task, which, with all due respect, appeared in the proceedings of a Cognitive Science meetings, disregarding the considerable literature documenting their understanding many probability concepts, the relations between sample and populations, and many other concepts that go well beyond probability matching strategies. I feel that the reference to a single, not particularly crucial paper (6 quotes in Google Scholar) and disregarding the full literature about probabilistic abilities, is not particularly convincing. Again, it seems like the authors are making some sort of rhetorical point rather than making a substantial argument. I would reconsider this

full paragraph, which is going to raise many eyebrows.

After carefully reading R2's comment, we realized that a strong focus on young infants was not necessary. Moreover, it was not our intention to make a "strawman" theory here. To make sure that this was avoided, we removed both the reference to the A not B task and to the probability-matching study. As reported in a previous comment (point 1.1), we now mostly focus on older infants, and on studies that focus on probabilistic inferences and/or information-seeking, including the study by Gweon & Schulz (2011), which is closer in age (16-month-olds) and topic (i.e., behavioral flexibility) to our sample and research question.

9. -- l. 63: "Both toddlers and adults performed the task successfully, correctly identifying the location of the hidden animal in both conditions". Where is the statistical/graphical support for this statement? If it is so important as to open the result section, it should be reported in the main text and not in the supp mat. The statement is repeated in the discussion (l.131: "We found that active information search is already successful at 18 months of age, as indicated by toddlers' ability to find the target by exploiting the informative cue."

-- Supp mat: it is missing for what I can see, and so I could not examine it.

We apologize for the mistake in the previous version of the manuscript. There, we referred to the supplementary materials, while we intended to refer to the methods section. In the methods, there is a specific subsection (i.e., Performance) that focuses on these results. The reason we did not focus on this in the main results is that this section is quite long, and it would disrupt the flow of the manuscript, which transitions from the introduction to the results maintaining the focus on adaptive information seeking. We hope that the current structure allows us to still disclose in the methods all the results about participants' success in the task, while keeping the focus on adaptiveness.

10. -- l.124: ""[...] we demonstrated that this ability ... emerges earlier ... than previously demonstrated." Maybe there is a better phrasing.

As suggested, we have now changed the sentence to: "we demonstrated that this ability to seek information adaptively emerges much earlier in development than previously thought".

-- l. 134: "Although previous studies identified rudimentary forms of active learning from early infancy": this is a very generic statement, from which I personally cannot gather what the authors have in mind. At least, they should provide some direct references to some scientific articles (as opposed to a general encyclopedia entry), so that the reader understands what the authors have in mind, what these demonstrations are and why they are rudimentary.

We have improved the citations to this statement, reconnecting to the citations we mentioned in the introduction. We also changed the word "rudimentary" to "simpler".

11. -- l. 139: what is ref 23? published? unpublished? One cannot say from the quoted ref.

The paper cited is now published. We updated the reference accordingly (now ref. 32).

12. -- (l. 62 ff): Fig 1 is Fig 2, clearly.

We have corrected the figure numbers throughout the manuscript.

13. -- l 154-5 : Typos. You mean a sample of 59 toddlers and of 28 adults?

We are not sure what appeared in R2's document due to the conversion in format file made by the journal. We intended to report that the power analysis estimated a required sample of toddlers ≥ 59 , and a sample of adults ≥ 28 .

14. -- in the osf storage I found the data but not the analysis files, so I could not check the analyses.

We apologize for the mistake. We did not include those materials among the ones that can be disclosed by the reviewers. We have now made everything open access at the following link: <https://osf.io/rfx5u/>.

15. -- (l. 315) the number of the quotes 7 and 10 are missing in my version of the ms (pdf). Also, in the pdf version Figure 3 is severely degraded, half-eaten up, probably because of the pdf conversion on the site. I had to go to the original version to see it.

We contacted the editorial team to make sure citations and picture quality will be preserved in the future.

Reviewer #3 (Remarks to the Author):

This is a potentially very interesting paper, showing that toddlers, from age at least aged 24 months onward alter their visual exploration behavior based on the nature of a search task.

I found the paper to be well written and thorough in its coverage of past literature, and I greatly appreciated the comparisons to optimal search generated by HMMs.

There was one aspect of the paper that I had a hard time following that I think matters critically for the nature of toddlers' performance on the task and for the novelty of these findings. Save the darkening screen, I had a hard discerning whether the training trials were different from the test trials, and critically, whether training differed across conditions. My central concern here is that if training was different across the conditions (i.e., the animal always appeared in one column for the uniform condition, but appeared in multiple columns across different trials in the skewed condition), but identical to the structure of the task on test, the findings may simply reflect a conditioning process (i.e., toddlers are rewarded during training for scanning broadly in the uniform than skewed condition; these behaviors then generalize to test trials). Essentially, more information about the similarity/dissimilarity between training and test trials is necessary to determine the novelty and ultimate contribution of the findings.

We thank the reviewer for pointing out this important aspect of the familiarization procedure. We have now addressed this difference between familiarization and test in the methods section. We believe that the familiarization procedure only critically differs in the statistical structure of the task across the two conditions (Skewed vs Uniform). Given that the screen was always entirely visible during familiarization, in both conditions it was possible to track the location of the target animal without engaging in scanning broadly across different locations. If anything, this works against our hypothesis that infants should scan more in the Uniform condition, because during training they learn that scanning is not required to find the target. Only when they reach the test phase of the Uniform condition, they have to realize that they cannot rely on just tracking the animal, and have to resort to an alternative strategy (i.e., scan for the informative cue). To clarify this point, we have added the following paragraph to the methods section (p. 7, lines 7-12):

“During familiarization trials, the screen was always visible, although it became increasingly darker. Thus, it was always possible to track the location of the target animal without the need to rely on the informative cue. However, during test trials, where the screen was completely dark, participants could not rely on tracking the animal anymore. In the Skewed condition, they could use the previously-observed evidence to predict its most likely location. In the Uniform condition, finding the animal required scanning across cue locations.”

In addition, we randomized the order of presentation of the two conditions between subjects. Half of the participants were first exposed to two Uniform blocks, and then transitioned to the Skewed blocks, while the other half of the participants had the opposite exposure (Skewed first, Uniform next). We reasoned that if the different familiarization was affecting the results by conditioning different behaviors, we should have observed a difference in the number of scanning movements depending on which condition participants were first exposed to. Specifically, we should expect more scanning movements if the first condition was the Uniform one. However, this effect was not significant (beta = -0.26, SE = 0.946613, t = -0.27, p = 0.79).

In summary, we believe that differences in familiarization cannot have caused the different patterns of exploration that we observe at test.

Reviewers' comments:

Reviewer #1 (Remarks to the Author):

The authors' revisions are helpful, particularly in their expanded framing and literature review. The terminology and description associated with the model are also now more appropriate. As the authors note, however, "all other aspects of the model remain intact". My initial review of the paper included the following paragraph, which the authors omitted in their response:

"In conclusion, I think the manuscript (and the model) needs substantial work before it is ready for publication in a high profile journal. The task and results, while motivated by interesting and important questions, are not by themselves sufficiently interesting to sustain the paper, and I think it needs a rewrite with more theoretical motivation and more explanation of why the results are interesting and informative. If the model is intended to be a significant part of the story, it needs substantial work as well; if it is intended to be an ideal observer analysis in that tradition, as advertised by the "optimal" terminology, the authors need to demonstrate that it has that property. These are major revisions."

I find that the theoretical motivation has been well addressed and the results better explained and put into context. The presented model is described more accurately (and I appreciate replacing "optimal" with "efficient" or "effective"), but as it is unchanged in substance, I still think that it makes a minor contribution. To develop a really satisfactory model would require a different modeling approach; I take it that the authors understood this from my comments and chose rather to improve how they presented the model they already had. I'm disappointed in this regard, but leave it to the editor to decide how to proceed on that front.

- Figure 2D is appreciated, but I found the style confusing. It seems more confusing than helpful to me to show (jittered) binary data, and I had to think carefully to understand the panel. The authors should also consider adding indicators (stars, etc) to show significance tests between conditions - its not obvious from the error bars that conditions are different.

Reviewer #2 (Remarks to the Author):

This version of the paper clarifies several issues and is an improvement with respect to the previous version. I was positive on the first version, I am positive on this version as well.

I am satisfied by most of the changes and the explanations of the points I raised. Still, in the revision I perceive a tendency to over-stress the novelty of the result which does not do justice to existing literature. Definitely the work of Schulz, Feigenson and their collaborators, as well as others, does show that toddlers flexibly adapt their problem-solving strategies to the information available, and are "tailored to the characteristics of the environment they are presented with" -- how else to describe their results?

So I am a bit puzzled by the suggestions that the present manuscript says something that was not said in previous work, in this respect.

Rather, the authors could better insist on the specific methodological advances of the paper, which to me are given by a fantastic experimental paradigm to explore visual search and study how different statistical relations in the environment can selectively guide infants' exploratory processes. For this, I believe they make a good case for the novelty of the work.

I think it's important to make an effort to locate the real novelty of the paper by doing justice to the work that has been done already. This will help the authors to be sharper in defining what they did, and will shield them from easy criticisms of it.

Here some other comments towards a potentially improved final version.

p 1 l. 18 : "Adaptive behavior is the hallmark of human intelligence". Well, this is highly questionable. It seems to me at best an unwarranted hyperbole. Adaptive behavior is the hallmark of most living beings. What is specifically human here? I suggest the authors find another way to convince the reader of the importance of their paper, there will be those who stop reading here.

P 1 l, 26 "These results show that toddlers' search strategies are more sophisticated than previously thought,". Previously thought by whom? And previously when? The same phrase is in pp 4-5, with reference to Ruggeri et al. (2019), but in that paper (if I am not wrong) there is no statement that the authors think that toddlers should not be able to display adaptive behavior in visual search, or for that matter, in general. It's only that younger children were not tested. I suggest to avoid these blank statements unless they are important for the argument -- in which case they should be supported.

p. 1 40-1: "Evidence of the ability to actively devise flexible information-seeking strategies is lacking in children below the age of 3". Ditto. This formulation does not do justice of previous work. There is plenty of literature that has been published dedicated to tasks that can be described as studies on information-seeking strategies, some of which are quoted in this version of the ms.

Example 1:

p. 2 1-4 "whether they search for information independently and unprompted," Isn't what Stahl & Feigenson show? And also others?

Example 2:

"and whether they change their search strategy depending on the specific probabilistic structure of the environment.": Isn't what Gweon & Schulz showed? In their response, the authors argue:

"In our study, infants need to devise actions to gather information (i.e., find the informative cue) with the goal to constrain their search for information (i.e., finding the location of the target animal). In Gweon and Schulz (2011), actions are devised to maximize their success to operate the toy, rather than to gather information by devising a specific information-seeking strategy."

But I am not convinced that this is a fair representation of what the work by Gweon & Schulz show. In their experiments, toddlers select which experimenter to trust on the basis of the series of successes/failures in providing reliable information. This is changing search strategy (search for information) on the basis of the probabilistic structure of the environment, for what I can tell. The fact that toddlers want to operate the toy is exactly equivalent to the fact that in the task of the authors' manuscripts toddlers want to find the animal -- these aspects are irrelevant to the process by which toddlers' goals is achieved. And for what I can see, in both the current work and Gweon's work the process requires adaptation to the statistical distributions in the environment (the success/failures of the experimenters in the case of Gweon, the validity of the cue in the case of the current experiment).

I also could make the same observation for other research lines, such as for example the work by Linda Smith, or Xu, and their collaborators. I think that the authors should find a better way to specify why their paper is really original in the face of previous work.

The bottom line is: they don't need to show that *ALL* what they do is novel and original; they have enough good results, and present such an elegant paradigm, that only on the basis of it I would accept publication. Also the discussion about the relation between the most efficient behavior, as derived by the model, and what adults, young toddlers and old toddlers do, is novel and worth stressing. All these are sufficient advances for the paper to have a general import. But be fair to the previous work, please.

P. 2 | 15-26: I wonder whether the extensive presentation of the children's' results is adding anything to the current paper, which is not probing anything related to constraint vs hypothesis-probing. I find this par distractive; perhaps the authors can make their point by simply recalling the theoretical claim they make, i.e., that there is evidence for adaptive competence in children but not in toddlers (but see my doubts above)

Reviewer #3 (Remarks to the Author):

The authors have provided a convincing argument for why their results cannot be explained by a conditioning process during familiarization. I am satisfied with their response.

Reviewers' comments

Reviewer #1 (Remarks to the Author):

The authors' revisions are helpful, particularly in their expanded framing and literature review. The terminology and description associated with the model are also now more appropriate. As the authors note, however, "all other aspects of the model remain intact". My initial review of the paper included the following paragraph, which the authors omitted in their response:

"In conclusion, I think the manuscript (and the model) needs substantial work before it is ready for publication in a high profile journal. The task and results, while motivated by interesting and important questions, are not by themselves sufficiently interesting to sustain the paper, and I think it needs a rewrite with more theoretical motivation and more explanation of why the results are interesting and informative. If the model is intended to be a significant part of the story, it needs substantial work as well; if it is intended to be an ideal observer analysis in that tradition, as advertised by the "optimal" terminology, the authors need to demonstrate that it has that property. These are major revisions."

I find that the theoretical motivation has been well addressed and the results better explained and put into context. The presented model is described more accurately (and I appreciate replacing "optimal" with "efficient" or "effective"), but as it is unchanged in substance, I still think that it makes a minor contribution. To develop a really satisfactory model would require a different modeling approach; I take it that the authors understood this from my comments and chose rather to improve how they presented the model they already had. I'm disappointed in this regard, but leave it to the editor to decide how to proceed on that front.

We apologize for not including the reviewer's concluding summary in our response. However, we would like to clarify how we believe we have addressed all the points summarized in this final paragraph when responding to the individual points raised by the reviewer.

We are glad to read that the reviewer is satisfied by our responses to their initial concerns about our papers' theoretical motivations. We would like to clarify a few points further, with the aim of overcoming Reviewer 1's remaining concerns. In particular, we understand that our changes to the framing of the model were not appreciated by the reviewer, who was hoping for a different modeling approach. As the reviewer said previously, "*The real issue here is with the derivation/specification of the "optimal strategy"*". To address this point, we have now implemented an *ideal learner model*, as previously suggested by the reviewer. Specifically, we removed all the previous computations that were used to quantify the most efficient exploration patterns in the Skewed and Uniform conditions. Instead, we implemented an ideal reinforcement-learning model that could move across the different states (i.e., locations) of the task to find the reward (i.e., the target animal). The model is now mentioned in the introduction (p. 2, lines 39-41):

"Specifically, participants in the Uniform condition engaged in more visual scanning to find the informative cue, and devised more complex search patterns (as predicted by an ideal reinforcement-learning model, see Results)"

In the results, we dedicated a specific subsection to the model (p. 3, lines 34-45):

"Simulating efficient search. To test whether participants flexibly adjusted their exploration across conditions, we pitted participants' exploratory eye-movements against the search

strategies that were most efficient in the two conditions. We used an ideal reinforcement learning model to infer the most efficient search for the Skewed and for the Uniform condition. These simulations (see Methods section and Fig. 3C) indicated that more predictable environments (i.e., the Skewed condition) are easier to learn and should result in simpler patterns of information search, as the location of the target is fully predictable without the aid of the cue. Conversely, less predictable environments (i.e., the Uniform condition) call for more complex patterns of information search, because they require to engage in a broader exploration directed at finding the cue, required to make the decision. We also compared participants' performance to a random search pattern, which consists of eye-movement patterns that are independent of the cue and the target locations. Hence, they result in random transitions from any location to any other in a completely unpredictable manner, thus corresponding to the highest possible level of complexity."

Figure 3. Results from the Markov models. **A.** The Markov model computes the probability of transitioning from each node (i.e., the 8 locations on the screen) to any other node. **B.** The transition matrix is the output of the Markov model, after observing the data. The data was divided by condition (Skewed vs. Uniform) and age (younger toddlers, older toddlers, and

adults), resulting in six transition matrices. A measure of entropy was computed for each matrix, thus quantifying the complexity of the exploratory patterns. **C.** We built an ideal reinforcement-learning model that learned to find the target animal in the Skewed and Uniform conditions. In the Skewed condition, the model learned that locations close in space to the target location (e.g., locations 0 and 5, when the target appears in location 4) are valuable, as they directly lead to the target. In the Uniform condition, the model learned that the cue locations (in purple) are valuable, as they might contain information that leads to the target. **D.** Complexity was higher for the Uniform compared to the Skewed condition for adults and older toddlers, but not for younger toddlers.

We also explained the model in detail in the methods section (p.9 lines 13-37):

“Ideal Reinforcement-Learning model

We used an ideal reinforcement-learning model to simulate the most efficient search patterns in the Skewed and Uniform conditions. We define S as the set of 8 discrete spatial states on the screen (i.e., the locations, or AOIs), with the particular neighborhood structure reflecting the spatial configuration (Fig. 3A). Specifically, we denote $S = \{0, 1, 2, 3, 4, 5, 6, 7\}$. From each state s in S , the set of available actions $A(s)$ corresponds to moving to a neighboring state. The neighbors for each state are predefined based on the spatial configuration.

The learning process is modeled through Q-Learning, utilizing a 3-dimensional Q-table Q , indexed by s, a, c where s is the state, a is the action, and c is the cue to the target. Assuming that the model is motivated by finding the target (i.e., the target is rewarding), the update rule for Q is:

$$Q(s, a, c) = Q(s, a, c) + \alpha \left[r + \gamma \max_{a'} Q(s', a', c) - Q(s, a, c) \right]$$

Here r is whether the target (i.e., reward) is observed, α is the learning rate, and γ is the discount factor. The learning rate α was set to 0.1, and the discount factor γ was set to 0.9 for the simulations. After taking an action, the agent transitions to a new state s' . The term $\max_{a'} Q(s', a', c)$ represents the maximum Q-value for the next state s' , considering all possible actions a' . The reward r is received based on the following conditions:

- *In the Skewed condition, $r = 1$ when the agent transitions to the predetermined target state, and $r = 0$ otherwise.*
- *In the Uniform condition, $r = 1$ only if the agent visits the cue state c before transitioning to the target state. If this condition is not met, $r = 0$.*

The model is trained over n episodes, with $n = 300$ for the simulations. For each episode, the state s , action a , and cue c are initialized randomly from their respective domains. We extracted the states visited by the model over all episodes for both the Skewed and Uniform condition. These sequences of states indicate the ideal model's exploration patterns, which are then compared to the participants' exploration patterns via Markov models.”

We would like to add that in Q-learning there is not a traditional "loss function" that is explicitly minimized, as one would encounter in supervised-learning models. However, the update rule for Q-learning effectively aims to reduce the temporal difference (TD) error, which can be viewed as a form of loss.

This new modelling approach led to additional insights compared to the previous model, as it shows that not only toddlers, but also adults are suboptimal compared to the ideal learner. This is now stated in the results section (p. 4, lines 17-25):

“We compared participants’ performance to the most efficient and random search patterns (Fig 3D). For the Skewed condition, we found that the exploratory patterns of both adults and older toddlers were more complex than required by the most efficient search (Ideal model log-entropy = 2.34; Adults: Mean log-entropy = 2.413, 95% CI = [2.403, 2.423]; Toddlers: Mean log-entropy = 2.455, 95% CI = [2.446, 2.465]). Similarly, for the Uniform condition, the complexity of the exploratory pattern of adults and older toddlers were more complex than required by the most efficient search (Ideal model log-entropy: 2.46, Adults: Mean log-entropy = 2.485, 95% CI = [2.475, 2.495], Toddlers: Mean log-entropy = 2.524, 95% CI = [2.514, 2.534]). In all cases, participants’ patterns were far more systematic than random search (log-entropy = 2.77).”

- Figure 2D is appreciated, but I found the style confusing. It seems more confusing than helpful to me to show (jittered) binary data, and I had to think carefully to understand the panel. The authors should also consider adding indicators (stars, etc) to show significance tests between conditions - its not obvious from the error bars that conditions are different.

We understand that, although highlighting these results was appreciated, the specifics of the figure were confusing. We have now moved these results in an independent figure (Fig. 4), including an intermediate step illustrating whether the transitions to cue and target locations were more frequent for adults or for toddlers (Fig. 4B). Moreover, in figure 4C, instead of showing error bars on binary data, we now show the proportion of transitions towards the cues, which make it clearer that toddlers are more likely than adults to make transitions to cue locations. Finally, we add an asterisk to indicate significance.

Figure 4. A. Comparison between the transition matrices of adults and infants, where red indicates transitions (i.e., eye movements from one location to the other) that have been made by infants more, and green indicates transitions that have been made by adults more. **B.** The overall frequencies of transitions that were more likely to be performed by infants and adults. **C.** A logistic regression model showed that adults were more exploitative (i.e., more likely to make a transition to a target area), while infants were more exploratory (i.e., more likely to make a transition to the cue areas).

We would like to stress that this figure is complementary to Figure 3. The reviewer has previously noted that our model is “unusual in that it doesn't assess the overall quality of participant

strategies, only their diversity -- while this is perhaps correlated with quality, it's not the same thing". Importantly, Figure 4 directly addresses these limitations, as it indicates that the observed patterns of results are qualitatively different in toddlers vs. adults, beyond differences in complexity. We have made this clearer also in the Results section (p. 4, lines 34-38):

"Finally, we tested for qualitative differences between adults' and toddlers' search patterns. Specifically, we analyzed the differences in transitional probability matrices between adults and toddlers with a logistic regression (Fig. 4). This allowed us to show that toddlers displayed an increased exploration of the cue locations compared to adults ($z = 2.86$, $\beta = 1.66$, $SE = 0.58$, $p = 0.004$), indicating an enhanced tendency to seek information."

In summary, we have now changed our modeling approach, by introducing an ideal reinforcement-learning model to compute the most efficient exploration patterns in the Skewed and Uniform condition. This approach offers a more formal definition of what the "most efficient" search strategies are, and a clearer optimality benchmark. In addition, we gave more weight to the qualitative differences between toddlers and adults, which are now depicted in Figure 4. We hope this modeling approach meets the expectations of the reviewer, and we are willing to make additional improvements if the reviewer thinks other changes might be helpful.

Reviewer #2 (Remarks to the Author):

This version of the paper clarifies several issues and is an improvement with respect to the previous version. I was positive on the first version, I am positive on this version as well.

I am satisfied by most of the changes and the explanations of the points I raised. Still, in the revision I perceive a tendency to over-stress the novelty of the result which does not do justice to existing literature. Definitely the work of Schulz, Feigenson and their collaborators, as well as others, does show that toddlers flexibly adapt their problem-solving strategies to the information available, and are "tailored to the characteristics of the environment they are presented with" -- how else to describe their results?

So I am a bit puzzled by the suggestions that the present manuscript says something that was not said in previous work, in this respect.

Rather, the authors could better insist on the specific methodological advances of the paper, which to me are given by a fantastic experimental paradigm to explore visual search and study how different statistical relations in the environment can selectively guide infants' exploratory processes. For this, I believe they make a good case for the novelty of the work.

I think it's important to make an effort to locate the real novelty of the paper by doing justice to the work that has been done already. This will help the authors to be sharper in defining what they did, and will shield them from easy criticisms of it.

We understand that the reviewer was happy with most of the changes, but would still want the previous related literature to be reviewed more broadly and more objectively. We now do this in the second paragraph of the Introduction. Instead of stressing the novelty of our findings, and then move on to contextualize them within the existing literature, we first address previous literature. We now mention more extensively the studies that documented early signs of adaptive behavior in toddlers. However, for completeness, we also mention the studies—suggested by the reviewer #2—

which show toddlers' failure in tasks that require adaptability. Only after this introduction of the literature, we point out that previous studies introduced toddlers to either one of two conditions, and then tested whether their behavior differed in the two conditions. However, our novel paradigm allows also for testing this ability within subjects. This is not trivial, as true flexibility would require toddlers to adapt strategy on the go, which has not been tested before. We now state the following (p. 1, line 39 – p. 2, line 12):

“Current evidence on whether adaptive information search is already present from early in life is conflicting. Some studies have shown that young children cannot optimally select the best exploration strategy when multiple options are available^{13,14}. For example, when given a clue that could narrow their search (e.g., an empty cup), toddlers do not preferentially choose the one cup that offers a guaranteed reward¹⁴. However, other studies have documented early signs of adaptive search in infants and toddlers. For example, infants are more likely to solicit information from a knowledgeable adult compared to an ignorant or unreliable one^{15,16}. Similarly, they rely on social partners when presented with cognitively demanding tasks, but tackle them on their own otherwise¹⁷. After failing to activate a toy, 16-month-old infants who were made to believe that the failure was due to their own inability were more likely to seek for help, while infants who were made to believe that the toy was malfunctioning were more likely to test the same behavior on other toys¹⁸. Relatedly, after observing an object unexpectedly pass through a wall, infants engaged in behaviors directed at testing their solidity (i.e., banging the object on a table)¹⁹. Taken together, these studies show that early search behavior is not rigid, as different environments elicit different responses. However, these between-subjects designs do not examine whether toddlers can dynamically adapt to changes in the environment, flexibly switching between different search strategies depending on the specific characteristics of the task at hand.”

Moreover, we now try to stress more the novelty of the paradigm we developed (p. 2, lines 20-23):

“In this paper, we introduce a novel experimental paradigm to investigate how toddlers adapt their exploration strategies to the characteristics of given environments. We devised a gaze-contingent eye-tracking task that allowed toddlers between 18 and 36 months of age, as well as adults, to actively and dynamically explore the environment presented on the screen (Fig. 1A).”

To summarize, we did relax the strength of some of our novelty claims, as requested by the reviewer. However, we believe that some key aspects of adaptive behavior in toddlers have been unaddressed by previous literature, and that our method offers a novel approach that is able to resolve them.

Here some other comments towards a potentially improved final version.

p 1 l. 18 : "Adaptive behavior is the hallmark of human intelligence". Well, this is highly questionable. It seems to me at best an unwarranted hyperbole. Adaptive behavior is the hallmark of most living beings. What is specifically human here? I suggest the authors find another way to convince the reader of the importance of their paper, there will be those who stop reading here.

We understand that this first sentence is especially important for convincing the reader of the importance of the paper. For this reason, we have now changed it to *“Adaptive information seeking is*

essential for humans to effectively navigate complex and dynamic environments”, which is more focused on why adaptive behavior is important.

P 1 l, 26 "These results show that toddlers’ search strategies are more sophisticated than previously thought,". Previously thought by whom? And previously when? The same phrase is in pp 4-5, with reference to Ruggeri et al. (2019), but in that paper (if I am not wrong) there is no statement that the authors think that toddlers should not be able to display adaptive behavior in visual search, or for that matter, in general. It's only that younger children were not tested. I suggest to avoid these blank statements unless they are important for the argument -- in which case they should be supported.

As suggested, we have changed the sentence to: “In the current work, we show how the ability to seek information adaptively emerges across the first years of life”.

p. 1 40-1: "Evidence of the ability to actively devise flexible information-seeking strategies is lacking in children below the age of 3". Ditto. This formulation does not do justice of previous work. There is plenty of literature that has been published dedicated to tasks that can be described as studies on information-seeking strategies, some of which are quoted in this version of the ms.

As suggested by the reviewer, we have now removed this sentence. We now first introduce previous evidence, and only then more specifically state that “these between-subjects designs do not examine whether toddlers can dynamically adapt to changes in the environment, flexibly switching between different search strategies depending on the specific characteristics of the task at hand.”

Example 1:

p. 2 1-4 "whether they search for information independently and unprompted," Isn't what Stahl & Feigenson show? And also others?

Although we disagree with Reviewer 2 on the interpretation of the results by Stahl & Feigenson (2015), we have now added this study to the introduction, and we have clarified the differences between their study and ours, as stated above.

Example 2:

"and whether they change their search strategy depending on the specific probabilistic structure of the environment.": Isn't what Gweon & Schulz showed? In their response, the authors argue:

"In our study, infants need to devise actions to gather information (i.e., find the informative cue) with the goal to constrain their search for information (i.e., finding the location of the target animal). In Gweon and Schulz (2011), actions are devised to maximize their success to operate the toy, rather than to gather information by devising a specific information-seeking strategy."

But I am not convinced that this is a fair representation of what the work by Gweon & Schulz show. In their experiments, toddlers select which experimenter to trust on the basis of the series of successes/failures in providing reliable information. This is changing search strategy (search for information) on the basis of the probabilistic structure of the environment, for what I can tell. The fact that toddlers want to operate the toy is exactly equivalent to the fact that in the task of the authors' manuscripts toddlers want to find the animal -- these aspects are irrelevant to the process by which

toddlers' goals is achieved. And for what I can see, in both the current work and Gweon's work the process requires adaptation to the statistical distributions in the environment (the success/failures of the experimenters in the case of Gweon, the validity of the cue in the case of the current experiment).

I also could make the same observation for other research lines, such as for example the work by Linda Smith, or Xu, and their collaborators. I think that the authors should find a better way to specify why their paper is really original in the face of previous work.

The bottom line is: they don't need to show that *ALL* what they do is novel and original; they have enough good results, and present such an elegant paradigm, that only on the basis of it I would accept publication. Also the discussion about the relation between the most efficient behavior, as derived by the model, and what adults, young toddlers and old toddlers do, is novel and worth stressing. All these are sufficient advances for the paper to have a general import. But be fair to the previous work, please.

Although we think that there is a difference between attempting to make a toy work (and relying on someone else when the attempts might be vain) an actively searching for information in different ways, we agree with the reviewer that this might not be the key aspect of the novelty of our study. We followed the reviewer's suggestion and insisted more on the methodological novelty. We are still convinced that the methodological novelty is not important in itself, but because it leads to key insights on toddlers' flexibility in searching for information.

As stated above, we now stress in the introduction that adaptiveness, in terms of flexibly switching between strategies on the go, is still a novel aspect that our study can address. In the discussion, we show that this helps to make sense of a wider literature, bridging studies on toddlers' cognitive flexibility and control to studies on toddlers' information search. Specifically, it is interesting to note that we did not find optimal behavior under 24 months, which provides a new element on the origins of cognitive flexibility. This might not be apparent if one considers previous studies, which have indeed shown 'local sensitivity' to correct information in infants (i.e., responding in different ways to different environments) but not 'global sensitivity' (i.e., switching adaptively on the go).

P. 2 | 15-26: I wonder whether the extensive presentation of the children's' results is adding anything to the current paper, which is not probing anything related to constraint vs hypothesis-probing. I find this par distractive; perhaps the authors can make their point by simply recalling the theoretical claim they make, i.e., that there is evidence for adaptive competence in children but not in toddlers (but see my doubts above)

We thank the reviewer for this comment. The reason we introduced this literature was to make clearer the idea of adaptation as "dynamic switching." However, we understand that introducing the difference between constraint-seeking vs hypothesis-probing questions is an extra step that may confuse the reader. At the same time, we appreciate that Reviewer 1 considered this paragraph as an improvement in the paper. For these reasons, we decided to keep the paragraph, but shorten it considerably, without mentioning the different types of questions. Instead, we focus on children's ability to adaptively switch between question types depending on task constraints, which more easily maps onto the "switching between exploratory strategies" of our task.

Reviewer #3 (Remarks to the Author):

The authors have provided a convincing argument for why their results cannot be explained by a conditioning process during familiarization. I am satisfied with their response.

REVIEWER COMMENTS

Reviewer #1 (Remarks to the Author):

The reinforcement learning model presented by the authors is much improved over the proposal from the previous submissions (which was not really a computational model as such). However, the current writeup (and potentially the research it describes) is unclear:

- Is the model intended to provide an account of how people *_act_* or how they *_learn to act_* (or both)? The optimal strategy for *_action_* in an environment is learnable through RL, but I don't believe that RL provides an account of *_optimal_* or *_ideal_* learning - if the others believe otherwise then I'd like to understand their reasoning. If the authors only intend to obtain the optimal policy, it's not clear to me why they chose reinforcement learning rather than MDP planning (the latter starts already knowing the details of the environment and solves for the optimal policy, which is appropriate because an ideal actor knows the environment already). The authors should clarify what job the model is supposed to do, and why their choices achieve it.

Further, on page 9 the authors "extract the states visited by the model over all episodes", which I think includes early exploration while the model is mostly unaware of the structure of the environment. If I'm understanding correctly, the paper thus compares humans with the behavior of a reinforcement learning model. If this is intended then that's okay, but again it's not an optimal learner, or at least I don't understand how it is. If the episodes extracted are from the learned policy, then that is potentially an ideal *_action_* model.

- In the Uniform condition, the RL environment is described as providing a reward when the agent visits the target state after first visiting the cue state. While this structure mirrors the human experiment, I don't understand how the contingency is encoded into the environment. A RL environment is a case of a MDP (*_Markov_* Decision Problem), which means that the reward that the agent gets from taking an action in a given state is independent of its previous history. So how does environment encode this contingency? There are methods, but the authors don't discuss their choice. At a minimum, I would like to know the details so that I can properly review this paper, but it would also be good to have a brief description in the paper of how this is handled for the reader interested in how the model works.

- Figure 4: the captions discuss infants while the graph legends mention toddlers. The word infant also shows up in Figure 2. Unless I misunderstood something, the data is collected from children 18-36 months (and adults), who are not referred to in this literature as infants.

To evaluate the overall paper, the most important point is to clarify what role the model is supposed to play, and how this goal is achieved. As mentioned, I don't think it can be described as an ideal learner. It may be described as an ideal *_actor_*. But it's not clear to me that this is what's intended. Discussion from the authors on all of these points is welcome and requested.

Reviewer #2 (Remarks to the Author):

In my previous reviews, I stressed the originality and interest of the novel experimental paradigm the authors introduce, but I had some reservations about the way previous literature was presented and discussed.

In this version, the authors made a very honest and convincing effort to address this shortcoming. They added a brief but fair treatment of previously done background work, and try to identify in what respects the current ms. is an advance. I find the current version quite balanced, and actually much

more useful to a general Nat Comm-like reader than the previous versions. While there are a few places where I would disagree with the authors' readings, by no means should these minor interpretive differences be an obstacle to the publication. Now the paper better exploits the very creative and original design they develop, and seems to me to open very solid lines of further research.

I do not comment on the changes to the model, which seem to me to be considerable in any case, and further improving the interest of the paper.

I am definitely in favor of its publication.

REVIEWER COMMENTS

Reviewer #1 (Remarks to the Author):

The reinforcement learning model presented by the authors is much improved over the proposal from the previous submissions (which was not really a computational model as such). However, the current writeup (and potentially the research it describes) is unclear:

1.1. Is the model intended to provide an account of how people *_act_* or how they *_learn to act_* (or both)? The optimal strategy for *_action_* in an environment is learnable through RL, but I don't believe that RL provides an account of *_optimal_* or *_ideal_* learning - if the others believe otherwise then I'd like to understand their reasoning. If the authors only intend to obtain the optimal policy, it's not clear to me why they chose reinforcement learning rather than MDP planning (the latter starts already knowing the details of the environment and solves for the optimal policy, which is appropriate because an ideal actor knows the environment already). The authors should clarify what job the model is supposed to do, and why their choices achieve it.

1.1. We thank the reviewer for the detailed evaluation of our model. Regarding whether the model is intended to provide an account of how people *_act_* or how they *_learn to act_*, we decided to use reinforcement learning to provide an account of how individuals learn to act over time, in environments where outcomes (rewards) are contingent on specific actions. Hence, we focus on the learning process and the development of action policies that maximize rewards, rather than on the instantaneous action choices themselves.

We now made this clear when the model is introduced in the results section, by stressing that we were interested in the learning aspect (p. 3, lines 34-38):

"To test whether participants learned to flexibly adjust their exploration across conditions, we pitted participants' exploratory eye-movements against the behavior of a reinforcement learning model. Specifically, in two sets of simulations, the model was introduced to either the Skewed or the Uniform condition, and learned how to efficiently search the target through trial and error."

Regarding whether the model provides an optimal account of learning, we agree with the nuanced distinction made by the reviewer. RL is primarily concerned with finding optimal or ideal *_action_* strategies rather than optimal learning processes per se. Although we are not comparing different learning algorithms, we believe we can still improve our modelling approach by asking: Within a simple RL model, which set of parameters values allows the model to perform best? Where by assessing the "best performance" we mean evaluating what exact model parameters values lead to find the target more quickly (i.e., with a lower number of steps taken on each episode). We now explain this in the methods section (p. 10, lines 1-10):

"To determine the values of α and γ that supported the most efficient search, we performed a grid-search algorithm across values of α (in the range from 0 to 1) and γ (in the range

from 0.5 to 1). Specifically, we trained each model over 300 episodes. For each episode, the state s , action a , and cue c are initialized randomly from their respective domains. After training, the Q -values were fixed, such that no additional updating was possible, and we tested the model's ability to identify the target on 1000 additional episodes. We evaluated the model performance in terms of number of steps required to identify the target, where a lower number of steps indicates the ability to successfully identify the target in a shorter period of time. The most efficient search was achieved by the model with parameter values $\alpha = 0.12$ and $\gamma = .83$, with a mean number of 6.7 steps taken to find the target, which was 3.4 steps lesser than the average performance (10.1 steps)."

The efficacy of this approach was confirmed by the agent's increasing success in the task, demonstrating that the selected RL framework can indeed capture the acquisition of an efficient action strategy.

In addition, we are now more careful with terminology. First, we removed all instances in which we refer to the model as ideal. Second, when we talk about the complexity (entropy) of the states visited by the model, we refer to it as "the complexity of the model's search patterns" instead of "ideal" complexity.

Finally, to address the difference between Markov Decision Processes (MDPs) and RL raised by the reviewer, we agree that this distinction is important, particularly in terms of how each method approaches the knowledge of the environment and the learning process leading to it. MDPs are used to compute the optimal policy given complete information (i.e., planning in a known environment). In cases where an "ideal actor" already knows the rules and outcomes of the environment, MDPs are a suitable choice for determining the optimal set of actions. However, MDPs do not model the learning process itself. To our knowledge, they are not designed to handle situations where an agent starts with incomplete knowledge of the environment and must learn this information through interaction (trial and error), which is what we aimed to capture here. As reported above, we now stress that our goal was to have a model that *learned* how to act (p. 3 in results section and p. 10 in the methods section).

1.2. Further, on page 9 the authors "extract the states visited by the model over all episodes", which I think includes early exploration while the model is mostly unaware of the structure of the environment. If I'm understanding correctly, the paper thus compares humans with the behavior of a reinforcement learning model. If this is intended then that's okay, but again it's not an optimal learner, or at least I don't understand how it is. If the episodes extracted are from the learned policy, then that is potentially an ideal `_action_` model.

1.2 As reported in the response point 1.1, we now state clearly that we compare human behavior with the behavior of a reinforcement-learning model. We hope that this clarification will help the reader to better understand the goal of our analysis.

2. In the Uniform condition, the RL environment is described as providing a reward when the agent visits the target state after first visiting the cue state. While this structure mirrors the human experiment, I don't understand how the contingency is encoded into the

environment. A RL environment is a case of a MDP (Markov Decision Problem), which means that the reward that the agent gets from taking an action in a given state is independent of its previous history. So how does environment encode this contingency? There are methods, but the authors don't discuss their choice. At a minimum, I would like to know the details so that I can properly review this paper, but it would also be good to have a brief description in the paper of how this is handled for the reader interested in how the model works.

2. We thank the reviewer for the insightful comment regarding the encoding of the contingency in our RL environment. Their point about maintaining the Markov property in a standard MDP framework is well-taken. In our current implementation, we have adopted an approach that, while deviating from a strict Markovian model, is tailored to the specifics of the task environment. At the implementation level, for each episode, a flag is initialized with value 0, and it remains 0 if the cue is not visited, but transitions to 1 if the cue is observed.

From the start, when the agent happens to visit the cue and then the target, it receives a reward, leading to higher Q-values for these actions. Over time, the Q-values reflect the utility of visiting the cue first because such actions consistently lead to rewards. Through these updates, the agent 'learns' to visit the cue first, even though it doesn't explicitly remember doing so. What is happening is that the Q-values for actions leading to the cue increase because those actions are part of sequences that lead to rewards.

By so doing, the model can learn the value of the cue location despite the apparent violation of the Markov property, enabling the agent to learn the value of the cue, even though the cue's importance is not explicitly encoded in the state representation. The agent effectively learns a policy that includes visiting the cue as a step toward obtaining the reward, even in the absence of an explicit memory of past actions. This outcome is a result of the nature of Q-learning, which is designed to find policies that maximize cumulative future rewards, even when those rewards are contingent on specific sequences of actions.

We acknowledge the importance of transparency in our modelling approach. In the revised manuscript, we included a detailed description of the model, specifically elaborating on how the cue flag is implemented within the learning algorithm and the rationale behind its use (p. 9, lines 27-42):

"The reward r is received based on the following conditions:

- In the Skewed condition, $r = 1$ when the agent transitions to the predetermined target state, and $r = 0$ otherwise.*
- In the Uniform condition, $r = 1$ only if the agent visits the cue state c before transitioning to the target state. If this condition is not met, $r = 0$. Hence, visiting the cue functions as a flag, determining whether the subsequent reward (i.e. observing the target) will be obtained when visiting the target state.*

Q-values are updated after each step based on the reward received and the expected future rewards (which are a function of the Q-values of the next state). The updating formula inherently accounts for the delayed nature of the reward because it propagates the value of future rewards back to earlier states. In the uniform condition, the agent has no initial

knowledge of the importance of the cue. However, through exploration, it occasionally visits the cue and then the target, receiving a reward. The Q-learning algorithm updates the Q-values to reflect this: the value of actions leading to the cue (when followed by actions leading to the target) increases. Over time, as the agent experiences more such successful sequences, the Q-values for actions leading to the cue state increase further, reflecting its importance in obtaining future rewards."

3. Figure 4: the captions discuss infants while the graph legends mention toddlers. The word infant also shows up in Figure 2. Unless I misunderstood something, the data is collected from children 18-36 months (and adults), who are not referred to in this literature as infants.

We thank the reviewer for spotting this oversight. Both figure captions have been corrected, now referring to the participants as "toddlers".

To evaluate the overall paper, the most important point is to clarify what role the model is supposed to play, and how this goal is achieved. As mentioned, I don't think it can be described as an ideal learner. It may be described as an ideal `_actor_`. But it's not clear to me that this is what's intended. Discussion from the authors on all of these points is welcome and requested.

We hope that our response and changes to the manuscript clarified that the model is intended to capture "learning how to act" instead of "action" only. However, we agree that the model is not an ideal learner in a strict sense, and to avoid misunderstandings we refrain from referring to it in these terms. At the same time, we also think that the model is not just an ideal actor, as we look at its "search patters" (i.e., the visited states) also while learning is happening and the environmental structure is still unknown. Hence, we simply refer to it as a reinforcement-learning model. To improve our modelling approach, we introduced a formal assessment of what sets of parameters would lead to the best performance, indicated in terms of fastest path to identify the target. We believe that in the current version we are more careful with terminology and more explicit about the exact implementation of our model (i.e., how observing the cue affects learning), and look forward to hearing from the reviewer if any aspect remains unclear, or if there is disagreement on any of our choices.

Reviewer #2 (Remarks to the Author):

In my previous reviews, I stressed the originality and interest of the novel experimental paradigm the authors introduce, but I had some reservations about the way previous literature was presented and discussed.

In this version, the authors made a very honest and convincing effort to address this shortcoming. They added a brief but fair treatment of previously done background work, and try to identify in what respects the current ms. is an advance. I find the current version quite balanced, and actually much more useful to a general Nat Comm-like reader than the previous versions. While there are a few places where I would disagree with the authors' readings, by no means should these minor interpretive differences be an obstacle to the publication. Now the paper better exploits the very creative and original design they develop, and seems to me to open very solid lines of further research.

I do not comment on the changes to the model, which seem to me to be considerable in any case, and further improving the interest of the paper.

I am definitely in favor of its publication.

We appreciate the reviewer's recognition of our efforts to enhance the manuscript, particularly in addressing the literature presentation and discussion. The reviewer's comments have been instrumental in refining our work and making it more accessible to a broader audience. We acknowledge the minor interpretive differences they highlighted and will consider these perspectives in our future research.

REVIEWERS' COMMENTS

Reviewer #1 (Remarks to the Author):

The authors provided a thorough response, including a significantly more detailed model evaluation compared to the previous manuscript version. I am satisfied with the modeling approach and the writeup. I'm still not sure that the work is of exceptional interest, but the technical and expository aspects of the paper are ready for publication.

One optional suggestion to the authors - it would be good to briefly mention, somewhere, that they applied reinforcement learning even though the environment isn't strictly Markovian. Doing so is fine provided that claims about optimality are weakened (which the authors did), but it is non-standard, and a close reader might be confused or want more information, as I did after my initial reading.